# TARGET BEFORE YOU PERTURB:
# ENHANCING LOCALLY PRIVATE GRAPH LEARNING VIA TASK-ORIENTED PERTURBATION

## ABSTRACT

Graph neural networks (GNNs) have achieved remarkable success in graph representation learning and have been widely adopted across various domains. However, real-world graphs often contain sensitive personal information, such as user profiles in social networks, raising serious privacy concerns when applying GNNs to such data. Consequently, *locally private graph learning* has gained considerable attention. This framework leverages local differential privacy (LDP) to provide strong privacy guarantees for users' local data. Despite its promise, a key challenge remains: how to preserve high utility for downstream tasks (*e.g.*, node classification accuracy) while ensuring rigorous privacy protection. In this paper, we propose `TOGL`, a Task-Oriented Graph Learning framework that enhances utility under LDP constraints. Unlike prior approaches that blindly perturb all attributes, `TOGL` first targets task-relevant attributes before applying perturbation, enabling more informed and effective privacy mechanisms. It unfolds in three phases: *locally private feature perturbation*, *task-relevant attribute analysis*, and *task-oriented private learning*. This structured process enables `TOGL` to provide strict privacy protection while significantly improving the utility of graph learning. Extensive experiments on real-world datasets demonstrate that `TOGL` substantially outperforms existing methods in terms of privacy preservation and learning effectiveness.

## 1 INTRODUCTION

Graph Neural Networks (GNNs) (Wu et al., 2020; Kipf & Welling, 2017) have emerged as powerful tools for learning representations from graph-structured data, achieving remarkable success in diverse applications such as social network analysis (Li et al., 2019; Wu et al., 2022; Sankar et al., 2021), recommendation systems (Ying et al., 2018; Sharma et al., 2024), and bioinformatics (Fout et al., 2017; Bessadok et al., 2022). Despite these successes, applying GNNs to real-world graphs often involves processing sensitive user data, such as profiles and behavioral logs on social networks. This raises substantial privacy risks, as recent studies have shown that adversaries can exploit trained GNNs to recover private information (Zhang et al., 2022; Meng et al., 2023; Wang & Wang, 2022; Yuan et al., 2024; Zhang et al., 2024b). Therefore, it is imperative to design an efficient privacy-preserving GNN framework that protects users' private information throughout the learning process.

Recently, *locally private graph learning* (Sajadmanesh & Gatica-Perez, 2021; Lin et al., 2022; Pei et al., 2023; Li et al., 2024; He et al., 2025a) has garnered considerable attention from the security and privacy research community. In this framework (as illustrated in Fig. 1(a)), each user independently perturbs their original node features using a local differential privacy (LDP) (Dwork et al., 2006) mechanism. The privacy level is governed by a parameter $\epsilon$, known as the *privacy budget*, where a smaller $\epsilon$ implies stronger privacy guarantees. The perturbed node features are then transmitted to an untrusted third-party server, which performs private graph learning on the noisy data to support downstream tasks such as node classification (Kipf & Welling, 2017) and link prediction (Zhang & Chen, 2018). LDP ensures that even if the transmitted data is intercepted, an adversary cannot reliably infer an individual's true input, thereby providing strong privacy guarantees in the local setting.

However, striking a balance between privacy and utility remains a significant challenge. Users' node features are typically high-dimensional and composed of multiple distinct attributes. Given a total

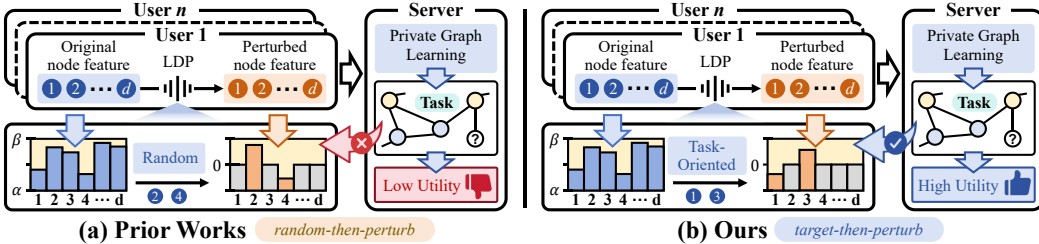

Figure 1: **Comparison of prior works with ours.** (a) Prior work adopts a "*random-then-perturb*" paradigm, where a random subset of attributes (*e.g.*, ❷ and ❹) is selected for perturbation while the rest are discarded (zeroed out), potentially overlooking task-relevant information. (b) In contrast, our work follows a "target-then-perturb" paradigm, which explicitly identifies task-relevant attributes (*e.g.*, ❶ and ❸) and perturbs them to enhance learning utility under the same privacy guarantee.

privacy budget $\epsilon$ (which is usually small), naïvely dividing $\epsilon$ equally across all feature dimensions and perturbing each leads to substantial utility loss. To mitigate this, as shown in Fig. 1(a), recent state-of-the-art approaches (Sajadmanesh & Gatica-Perez, 2021; Lin et al., 2022; Pei et al., 2023; Li et al., 2024; Jin & Chen, 2022; He et al., 2025a) adopt a "*random-then-perturb*" paradigm: a small subset of attributes is randomly selected, and the entire $\epsilon$ is evenly allocated among them, while the remaining unselected attributes are zeroed out. This strategy helps preserve the quality of the selected attributes by reducing the per-dimension noise under the same privacy guarantee (same $\epsilon$). Nevertheless, LDP mechanisms based on this paradigm still suffer from limited utility in practice.

In this paper, as illustrated in Fig. 1(b), we propose TOGL, a Task-Oriented Graph Learning framework designed to enhance utility under LDP constraints. TOGL is motivated by the realistic observation that, in realistic graph learning scenarios, the utility of downstream tasks often relies on only a small subset of attributes within the high-dimensional node features. For example, in a node classification task like credit risk prediction (Wang et al., 2021), the model may primarily rely on a few critical attributes (such as income, repayment history, and employment), while other attributes contribute little. Leveraging this insight, TOGL introduces a novel "*target-then-perturb*" paradigm for LDP. In contrast to existing methods that randomly select attributes to perturb, TOGL first targets task-relevant attributes and then applies perturbation specifically to them. This targeted perturbation strategy facilitates more accurate privacy preservation and achieves improved learning utility.

Achieving the above objective is highly non-trivial due to several key challenges. First, identifying task-relevant attributes under LDP constraints is inherently difficult, as the available data has already been perturbed, obscuring useful patterns. Second, the key attribute selection process itself must be privacy-preserving and must not weaken the privacy guarantees compared to existing methods, further complicating the extraction of informative task signals. Third, naively maximizing task relevance can be counterproductive: overemphasizing task-specific attributes may impair the graph's topological distinguishability, thereby degrading the model's ability to capture structural patterns essential for generalization. These challenges highlight the need for a careful balance among task utility, attribute privacy, and structural information, along with clear metrics to guide this trade-off.

To address these challenges, TOGL follows a three-phase pipeline: ① *Locally Private Feature Perturbation*. Each user perturbs their node features using an LDP mechanism and uploads the perturbed features to the server. The server denoises features via multi-hop aggregation to enable more accurate attribute analysis. ② *Task-Relevant Attribute Analysis*. We introduce two methods to identify task-relevant attributes: Fisher Discriminant Analysis (FDA), which captures discriminative signals across classes, and Sparse Model Attribution (SMA), which highlights sparse, high-impact features based on model behavior. ③ *Task-Oriented Private Learning*. The model selectively perturbs a combination of task-relevant attributes and randomly sampled attributes to balance task consistency and topological distinguishability. Throughout the pipeline, TOGL ensures strict end-to-end LDP guarantees while significantly enhancing the utility of private graph learning.

**Contributions.** The key contributions are as follows. ① We introduce a novel task-oriented perspective for studying locally private graph learning. ② We propose TOGL, a Task-Oriented Graph Learning framework that enhances utility while adhering to LDP constraints. ③ Extensive experiments on six real-world datasets demonstrate substantial utility improvements over existing baselines.

## 2 PRELIMINARIES

In this section, we first define the problem (Sec. 2.1), then present the essential background of local differential privacy (Sec. 2.2), and finally introduce the framework of locally private graph learning (Sec. 2.3). Important notations are summarized in Appendix A.

### 2.1 PROBLEM DEFINITION

Consider a graph $\mathcal{G} = (\mathcal{V}, \mathcal{E})$, where $\mathcal{V}$ is the set of nodes and $\mathcal{E}$ is the set of edges. Let $\mathbf{X} \in \mathbb{R}^{|\mathcal{V}| \times d}$ denote the node feature matrix, where each node $v \in \mathcal{V}$ is associated with a $d$-dimensional feature vector $\mathbf{x}_v \in [\alpha, \beta]^d$ containing sensitive user information[1]. Each node is also associated with a label $y_v \in \mathcal{Y}$, where $\mathcal{Y} = \{y_1, y_2 \cdots, y_C\}$ denotes the set of possible classes. The objective is to train a GNN model to perform tasks such as node classification (Kipf & Welling, 2017) using graph data. However, uploading such data to an untrusted server for GNN-based graph learning introduces significant privacy risks (Zhang et al., 2022; Meng et al., 2023; Zhang et al., 2024b; Wang & Wang, 2022; Yuan et al., 2024). To address this, *locally private graph learning* (Sajadmanesh & Gatica-Perez, 2021; Lin et al., 2022; Pei et al., 2023; Li et al., 2024; He et al., 2025a; Jin & Chen, 2022) seeks to leverage LDP (Yang et al., 2024) to protect individual node privacy while maintaining high utility (*e.g.*, node classification accuracy) in graph learning tasks. This paper proposes to optimize existing LDP perturbation mechanisms from a *task-oriented perspective*, thereby promoting a more utility-efficient framework for locally differentially private graph learning.

### 2.2 LOCAL DIFFERENTIAL PRIVACY

LDP (Yang et al., 2024) is a rigorous privacy framework that enables meaningful data analysis while protecting individual privacy. It has been widely adopted in various decentralized data collection and distribution settings (Duchi et al., 2013; Kairouz et al., 2014; 2016; Cormode et al., 2018; Wang et al., 2019a;b). By introducing randomized noise into the data processing pipeline, LDP offers strong privacy guarantees for users' raw data. Under the LDP paradigm, each user locally perturbs their data $x$ using a randomized mechanism $\mathcal{M}$ before transmitting it to potentially untrusted servers for downstream tasks. The mechanism $\mathcal{M}$ must satisfy the following definition:

**Definition 1** ($\epsilon$-LDP). *A randomized algorithm $\mathcal{M} : \mathcal{X} \to \mathcal{Z}$, where $\mathcal{X}$ is the domain of all input $x$, satisfies $\epsilon$-local differential privacy ($\epsilon$-LDP) if for any two inputs $x, x' \in \mathcal{X}$ and any output $z \in \mathcal{Z}$,*

$$\Pr[\mathcal{M}(x) = z] \leq e^{\epsilon} \cdot \Pr[\mathcal{M}(x') = z], \quad \epsilon > 0. \tag{1}$$

The parameter $\epsilon$, known as the *privacy budget*, quantifies the trade-off between privacy and utility. A smaller $\epsilon$ implies that $\mathcal{M}$ offers stronger privacy protection, but typically at the cost of reduced utility. The two most commonly studied properties (Dwork et al., 2014) of LDP are as follows:

**Theorem 1** (Sequential Composition). *If $\mathcal{M}_i : \mathcal{X} \mapsto \mathcal{Z}_i$ satisfies $\epsilon_i$-LDP for each $i \in \{1, 2, \ldots, n\}$, then the composed mechanism $\mathcal{M} = (\mathcal{M}_1, \mathcal{M}_2, \ldots, \mathcal{M}_n) : \mathcal{X} \mapsto \prod_{i=1}^{n} \mathcal{Z}_i$ satisfies $(\sum_{i=1}^{n} \epsilon_i)$-LDP.*

**Theorem 2** (Post-Processing Invariance). *Let $\mathcal{A} : \mathcal{X} \mapsto \mathcal{Z}$ satisfies $\epsilon$-LDP, and let $\mathcal{F} : \mathcal{Z} \mapsto \mathcal{Z}'$ be any (possibly randomized) mapping. Then the composed mechanism $\mathcal{A} \circ \mathcal{F} : \mathcal{X} \mapsto \mathcal{Z}'$ satisfies $\epsilon$-LDP.*

### 2.3 LOCALLY PRIVATE GRAPH LEARNING

Locally private graph learning consists of two steps: *perturb* and *learn*, where data perturbation is performed on the user side, and graph learning is carried out on the server side.

- *Perturb.* To protect the privacy of node features, three state-of-the-art LDP mechanisms have been proposed: the piecewise mechanism (PM) (Pei et al., 2023; Wang et al., 2019a), the multi-bit mechanism (MB) (Lin et al., 2022; Sajadmanesh & Gatica-Perez, 2021; Jin & Chen, 2022), and the square wave mechanism (SW) (Li et al., 2024; 2020). Given the high dimensionality of node features, blindly perturbing each attribute can significantly impair data utility. To mitigate this, these mechanisms adopt a "*random-then-perturb*" paradigm (Fig. 1(a)) to balance privacy with utility. The general ingestion process consists of two steps. First, $m$ values are randomly selected

---

[1]This paper focuses on protecting users' node features, which are often the most sensitive and critical for downstream learning tasks. Our approach is orthogonal to existing privacy-preserving techniques for neighbor lists (Hidano & Murakami, 2024; Zhu et al., 2023a), and can be seamlessly integrated with them. While we do not directly consider link-level privacy in this work, TOGL remains effective, and additional experiments in Appendix D.5 demonstrate that it maintains strong performance even when neighbor lists are locally perturbed.

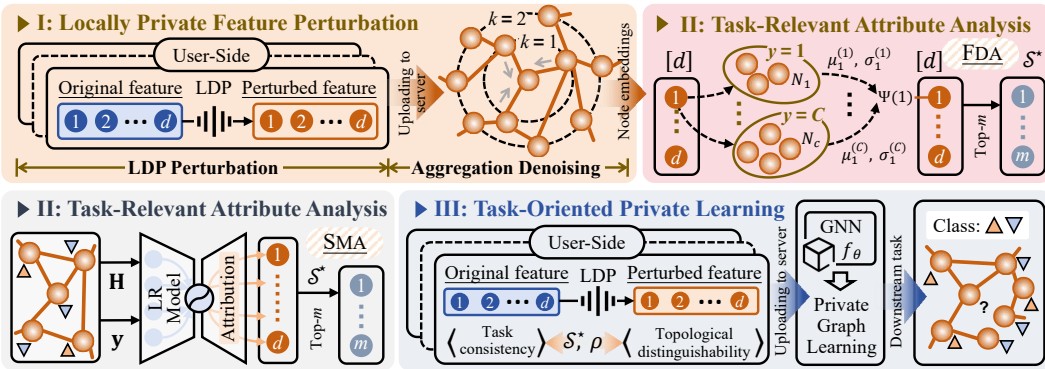

Figure 2: The overview of `TOGL` consists of three successive phases: locally private feature perturbation, task-relevant attribute analysis, and task-oriented private learning. Specifically, in Phase I, `TOGL` utilizes a LDP mechanism to collect initial node features and perform aggregation denoising, enabling subsequent attribute analysis. In Phase II, task-relevant attributes are scored using the FDA or SMA methods to identify the top-$m$ key attribute set $\mathcal{S}^\star$. Finally, in Phase III, `TOGL` conducts task-oriented private graph learning guided by $\mathcal{S}^\star$ to solve downstream tasks such as node classification.

without replacement from the $d$-dimensional feature space $[d]$. Then, an $\epsilon/m$-LDP perturbation is applied to each of the $m$ selected dimensions, while the remaining $d - m$ dimensions are zeroed. Specifically, for the `PM` in the one-dimensional setting (see App. B for more details on `MB` and `SW`), the input domain is $[\alpha, \beta]$, and the output range is $[-\mathcal{B}, \mathcal{B}]$, where $\mathcal{B} = \frac{e^{\epsilon/2}+1}{e^{\epsilon/2}-1}$. Given an original input value $x$, the perturbed value $x'$ is sampled from the following probability density function:

$$\Pr[x' = c|x] = \begin{cases} p, & \text{if } c \in [l(x), r(x)] \\ p/e^\epsilon, & \text{if } c \in [-\mathcal{B}, l(x)) \cup (r(x), \mathcal{B}] \end{cases}, \tag{2}$$

where $p = \frac{e^\epsilon - e^{\epsilon/2}}{2e^{\epsilon/2}+2}$, $l(x) = \frac{\mathcal{B}+1}{2} \cdot x - \frac{\mathcal{B}-1}{2}$, and $r(x) = l(x) + \mathcal{B} - 1$.

- *Learn.* After collecting all perturbed node features, the server performs private graph learning using a GNN model for downstream tasks such as node classification. The GNN iteratively updates node embeddings by aggregating local neighborhood information. At layer $k$, each node $v \in \mathcal{V}$ first aggregates the embeddings from its neighbors $\mathcal{N}(v)$, and then applies a learnable update function:

$$\mathbf{h}_{\mathcal{N}(v)}^k = \text{AGGREGATE}_k \left( \{ \mathbf{h}_u^{k-1} \mid u \in \mathcal{N}(v) \} \right), \quad \mathbf{h}_v^k = \text{UPDATE}_k \left( \mathbf{h}_{\mathcal{N}(v)}^k \right), \tag{3}$$

where $\mathbf{h}_u^{k-1}$ is the embedding of node $u \in \mathcal{N}(v)$ from the previous layer. The aggregation function AGGREGATE$(\cdot)$ (*e.g.*, mean, sum, or max) combines information from neighboring nodes, while the update function UPDATE$(\cdot)$ (*e.g.*, a neural network layer) refines the representation. The process is initialized with $\mathbf{h}_v^0 = \mathbf{x}_v$, where $\mathbf{x}_v$ denotes the input feature vector of node $v \in \mathcal{V}$.

## 3 METHODOLOGY

In this section, we detail the *Task-Oriented Graph Learning* (`TOGL`) framework under LDP constraints. `TOGL` is designed to optimize the initial LDP perturbation process for graph learning from a task-oriented perspective, thereby enhancing the utility of downstream learning tasks. Fig. 2 illustrates the overview of `TOGL`, and Alg. 2 presents the details.

### 3.1 LOCALLY PRIVATE FEATURE PERTURBATION

This phase begins with applying *LDP perturbation* to privately collect user node features, followed by *aggregation denoising* to enable attribute analysis in later phases.

**LDP Perturbation.** In `TOGL`, we aim to analyze the node features $\mathbf{x}_v \in [\alpha, \beta]^d$ of all users $v \in \mathcal{V}$ to identify a top-$m$ set of attributes, denoted as $\mathcal{S}^\star$, that are most relevant to the downstream learning task. However, under privacy constraints, direct access to the original node features is not available. Therefore, we first apply an LDP mechanism to perturb $\mathbf{x}_v$ and obtain its privatized version. As discussed in Sec. 2.3, we consider three state-of-the-art LDP mechanisms (`PM`, `MB`, and `SW`) suitable for protecting high-dimensional node features. These mechanisms are unified into a general LDP perturbation protocol $\Pi$, as presented in Alg. 1. The inputs to this protocol are $\mathbf{x}_v$ and $\epsilon$.

Specifically, in $\Pi$, the optimal perturbation size $m$ is determined based on the feature dimension $d$, coefficient $\delta$, and privacy budget $\epsilon$ (line 1). The coefficients are theoretically derived to minimize estimation error: $\delta = 2/5$ for PM, $5/11$ for MB, and $2/5$ for SW (please see Appendix B for details). Then, among the $d$ dimensions, a random subset $\mathcal{S} \subset [d]$ of the $m$ dimensions is selected for the perturbation, while the remaining dimensions are zeroed out (lines 2–7). Finally, the perturbed vector $\tilde{\mathbf{x}}_v$ is processed using the $\text{RECT}(\cdot)$ function to ensure unbiased estimation, *i.e.*, $\mathbb{E}[\mathbf{x}'_v] = \mathbf{x}_v$, which helps to further reduce the impact of noise (line 8). The final output is the perturbed node feature $\mathbf{x}'_v$.

---

**Algorithm 1** LDP Perturbation Protocol $\Pi$

---

**Input:** Node feature $\mathbf{x}_v \in [\alpha, \beta]^d$, privacy budget $\epsilon > 0$.
**Output:** Perturbed node feature $\mathbf{x}'_v \in [-\mathcal{B}, \mathcal{B}]^d$.
1: $m \leftarrow \max(1, \min(d, \lfloor \delta \cdot \epsilon \rfloor))$. ▷ *perturbation size*
2: Let $\tilde{\mathbf{x}}_v \leftarrow <0, 0, \cdots, 0>$.
3: Let $\mathcal{S} \subset [d]$ denote a subset of $m$ distinct dimensions selected uniformly at random without replacement from the index set $\{1, 2, \ldots, d\}$.
4: **for** *each sampled dimension* $i \in \mathcal{S}$ **do**
5:     Feed $\mathbf{x}_v[i]$ and $\epsilon/m$ as input to one-dimensional LDP mechanisms (*e.g.*, Eq. (2)), and obtain $\tilde{x}_{\text{tmp}}$.
6:     $\tilde{\mathbf{x}}_v[i] \leftarrow \tilde{x}_{\text{tmp}}$. ▷ *LDP perturbation*
7: **end for**
8: $\mathbf{x}'_v \leftarrow \text{RECT}(\tilde{\mathbf{x}}_v, \epsilon, m, d)$. ▷ *unbiased rectification*
9: **return** Perturbed node feature $\mathbf{x}'_v \in [-\mathcal{B}, \mathcal{B}]^d$.

---

**Aggregation Denoising.** Once all the perturbed node features $\mathbf{x}'_v$ are collected, the server proceeds with task-related attribute analysis. However, directly analyzing $\mathbf{x}'_v$ may yield suboptimal results due to substantial noise. To mitigate this issue, we consider the aggregation operation (Eq. (3)) and introduce the following proposition.

**Proposition 1.** *Let $\mathbf{h}_{\mathcal{N}(v)}$ denote the true aggregated embedding over the neighborhood $\mathcal{N}(v)$, and let $\overline{\mathbf{h}}_{\mathcal{N}(v)}$ be its locally perturbed counterpart. Then the discrepancy $\Upsilon$ between the two decays inversely with the size of the neighborhood, i.e., $\left\| \Upsilon \left\langle \overline{\mathbf{h}}_{\mathcal{N}(v)} - \mathbf{h}_{\mathcal{N}(v)} \right\rangle \right\| \propto 1/|\mathcal{N}(v)|$.*

Proposition 1 (see Appendix C.1 for details) indicates that increasing the neighborhood size $|\mathcal{N}(v)|$ reduces the discrepancy with the true aggregated embedding, thereby calibrating the error and improving subsequent attribute analysis. To this end, we perform $K$ recursive aggregation steps $\text{AGGREGATE}(\cdot)$ without applying any non-linear transformation, as defined below.

$$\overline{\mathbf{h}}^k_{\mathcal{N}(v)} = \text{AGGREGATE}_k(\{\overline{\mathbf{h}}^{k-1}_u \mid u \in \mathcal{N}(v)\}), \quad k \in \{1, 2, \cdots, K\}. \tag{4}$$

The process is initialized with $\overline{\mathbf{h}}^0_v = \mathbf{x}'_v$, and the final estimated node embedding is denoted by $\overline{\mathbf{h}}_v$.

### 3.2 TASK-RELEVANT ATTRIBUTE ANALYSIS

After completing Phase I and obtaining the estimated node embeddings for all nodes $v \in \mathcal{V}$ (denoted as $\mathbf{H} = \{\overline{\mathbf{h}}_1, \overline{\mathbf{h}}_2, \cdots, \overline{\mathbf{h}}_{|\mathcal{V}|}\}$), the server then utilizes them for key attribute analysis to identify the top-$m$ task-relevant attributes[2], denoted as $\mathcal{S}^\star$. In Phase II, we propose two task-relevant attribute analysis methods: *Fisher Discriminant Analysis* (FDA) and *Sparse Model Attribution* (SMA). FDA is grounded in classical statistical pattern recognition theory, evaluating each attribute independently based on its class separability (inter-class vs. intra-class variance). This makes FDA computationally efficient and robust when features are relatively independent. In contrast, SMA adopts a model-driven approach, using sparse logistic regression with $L_1$ regularization to capture task-adaptive feature dependencies. By learning which attributes jointly contribute to correct predictions, SMA can identify relevant features even when their importance arises from interactions rather than individual discriminative power. These two methods provide complementary perspectives: FDA offers statistical separability analysis, while SMA captures learned task-specific patterns.

**Fisher Discriminant Analysis (FDA).** Drawing inspiration from the classical Fisher discriminant criterion (Mika et al., 1999), we propose the FDA, which leverages statistical pattern recognition to quantify the task relevance of each individual attribute dimension. Specifically, given the aggregated embeddings $\mathbf{H} = \{\overline{\mathbf{h}}_v\}_{v \in \mathcal{V}}$ and their corresponding labels $\mathbf{y} = \{y_v\}_{v \in \mathcal{V}}$, we evaluate each dimension $j$ of the feature vector by measuring its ability to distinguish between different classes.

Formally, for each class $c \in \{1, 2, \ldots, C\}$, we compute the class-conditional mean $\mu_j^{(c)}$ and variance $\sigma_j^{(c)}$ of dimension $j \in \{1, 2, \ldots, d\}$, based on the set of embeddings $\overline{\mathbf{h}}_v$ where $y_v = c$. The overall

---

[2]The task-specific signals required by TOGL are broadly defined and not limited to explicit node labels. See Appendix F for a detailed discussion on their practical availability across different learning settings.

class-weighted mean $\bar{\mu}_j$ is given by: $\bar{\mu}_j = \sum_{c=1}^{C} (N_c/|\mathcal{V}|) \cdot \mu_j^{(c)}$, where $N_c$ is the number of nodes in class $c$ and $|\mathcal{V}|$ is the total number of nodes. The inter-class (between-class) variance and intra-class (within-class) variance for dimension $j$ are computed as:

$$S_B(j) = \sum_{c=1}^{C} \frac{N_c}{|\mathcal{V}|} (\mu_j^{(c)} - \bar{\mu}_j)^2, \quad S_W(j) = \sum_{c=1}^{C} \frac{N_c}{|\mathcal{V}|} \sigma_j^{(c)}. \tag{5}$$

We then define the Fisher discriminative score for the dimension $j$ as: $\Psi_{\text{FDA}}(j) = \frac{S_B(j)}{S_W(j) + \varsigma}$, where $\varsigma > 0$ is a small smoothing constant to ensure numerical stability. Intuitively, this criterion favors dimensions whose values are well-separated across classes (high between-class variance) and consistent within each class (low within-class variance). The top-$m$ dimensions with the highest $\Psi_{\text{FDA}}(j)$ scores are selected as candidates for task-relevant attributes, serving as the input to the final selection in Phase III (Sec. 3.3). This process is formalized in Eq. (6) and yields the final attribute set $\mathcal{S}^\star$.

$$\mathcal{S}^\star \leftarrow \text{TOP-K}_{j \in [d]} \Psi_{\text{FDA}}(j), \quad |\mathcal{S}^\star| = m. \tag{6}$$

**Sparse Model Attribution (SMA).** While FDA evaluates each attribute dimension independently based on class separability statistics, it does not capture potential interactions among features. As a complementary perspective, we consider a model-informed scoring strategy that reflects the joint contribution of attributes under a sparse predictive model. We train a multi-class logistic regression (LR) (LaValley, 2008) with $L_1$-regularization on $\mathbf{H} = \{\overline{\mathbf{h}}_v\}$ and $\mathbf{y} = \{y_v\}$. The objective is:

$$\mathbf{W}^* = \arg \min_{\mathbf{W} \in \mathbb{R}^{C \times d}} (1/|\mathcal{V}|) \cdot \sum_{i=1}^{|\mathcal{V}|} \ell(y_i, \mathbf{W}\mathbf{h}_i) + \lambda \|\mathbf{W}\|_1, \tag{7}$$

where $\ell$ is the softmax cross-entropy loss, and $\lambda > 0$ controls sparsity. After training, the attribution score for feature dimension $j$ is defined as: $\Psi_{\text{SMA}}(j) = (1/C) \sum_{c=1}^{C} |W_{c,j}^*|$. In Algorithm 2, Line 5, when SMA is selected for attribute analysis, we solve Eq. (7) to obtain $\mathbf{W}^*$, compute $\Psi_{\text{SMA}}(j)$ for all dimensions, and select the top-$m$ dimensions to form $\mathcal{S}^\star$. This procedure favors features that consistently contribute to correct predictions across classes, while suppressing redundant or noisy dimensions. Compared to FDA, it accounts for task-adaptive feature dependencies and can be more robust when class boundaries are non-linearly entangled.

### 3.3 TASK-ORIENTED PRIVATE LEARNING

After identifying the task-relevant attribute set $\mathcal{S}^\star$ via FDA or SMA, a new round of LDP perturbation is optimized under its guidance to enable fine-grained noise injection, thereby enhancing the private graph learning utility. However, the performance of graph learning is influenced by both task relevance and the level of topological

---

**Algorithm 2** Task-Oriented Graph Learning

**Input:** Graph $\mathcal{G}$, privacy budget $\epsilon$, LDP Protocol $\Pi$, hyperparameters $K$, $\rho$, etc.
**Output:** Trained GNN model $f_\Theta$.
/** `Phase I: Locally Private Feature Perturbation` **/
1: **for** each node $v \in \mathcal{V}$ in parallel **do**
2:    $\mathbf{x}_v' \leftarrow \Pi(\mathbf{x}_v, \epsilon/2)$.        $\triangleright$ See Alg. 1
3: **end for**
4: $\mathbf{H} \leftarrow \text{AGGREGATE}_K(\{\mathbf{x}_v'\})$.    $\triangleright$ Eq. (4)
/** `Phase II: Task-Relevant Attribute Analysis` **/
5: $\mathcal{S}^\star \leftarrow$ FDA or SMA.      $\triangleright$ See Sec. 3.2
/** `Phase III: Task-Oriented Private Learning` **/
6: **for** each node $v \in \mathcal{V}$ in parallel **do**
7:    Obtain $\mathcal{S}_v$ based on $\mathcal{S}^\star$ and $\rho$.    Eq. (8)
8:    $\mathbf{x}_v' \leftarrow \Pi(\mathbf{x}_v, \epsilon/2, \mathcal{S}_v)$.    $\triangleright$ See Alg. 1
9: **end for**
10: Train $f_\Theta$ on $\{\mathbf{x}_v'\}$ for downstream tasks.
11: **return** Trained GNN model $f_\Theta$.

---

distinguishability (Theorem 3). Directly setting the set $\mathcal{S}$ in Alg. 1 to $\mathcal{S}^\star$ ensures task consistency but undermines the topological smoothness of the graph (Corollary 1), diminishing the topology-aware noise calibration achieved via the message-passing of GNNs. To address this, given the global top-$m$ attribute set $\mathcal{S}^\star$, we define each node $v$'s personalized perturbation subset $\mathcal{S}_v$ as:

$$\mathcal{S}_v = \underbrace{\text{TOP-}m^\star(\mathcal{S}^\star)}_{\text{task-relevant}} \cup \underbrace{\text{RANDOMSAMPLE}([d] \setminus \mathcal{S}_{\text{fixed}}, m - m^\star)}_{\text{randomized diversity}}, \tag{8}$$

where $m^\star = \lfloor \rho \cdot m \rfloor$ and $\rho \in [0, 1]$ is a hyperparameter controlling the trade-off between *task consistency* and *topological distinguishability*, with $\mathcal{S}_{\text{fixed}}$ denoting the subset of attribute dimensions that are deterministically selected based on task relevance—that is, the top-$m^\star$ attributes in $\mathcal{S}^\star$.

**Complexity Analysis.** The computational complexity primarily arises from the $K$-hop neighborhood aggregation in Phase I and the attribute analysis (FDA or SMA) in Phase II. These cost $\mathcal{O}(K \cdot |\mathcal{E}| \cdot d)$ and $\mathcal{O}(|\mathcal{V}| \cdot d \cdot I + d \log d)$ respectively ($I$ is the number of iteration rounds of LR training), both scaling linearly with graph size and feature dimension. More details are provided in Appendix C.2.

## 4 THEORETICAL ANALYSIS

**Task–Topology Tradeoff.** As in Theorem 3 and Corollary 1, the utility of private graph learning is influenced not only by task consistency, but also by the preservation of topological distinguishability.

**Theorem 3.** *Let $f_\Theta : \mathbb{R}^d \to \mathbb{R}^C$ be a GCN-like node classifier trained under LDP-constrained inputs $\{\mathbf{x}'_v\}_{v \in \mathcal{V}}$, where each node $v \in \mathcal{V}$ perturbs only a fixed dimension subset $\mathcal{S} \subset [d]$. Then the upper bound of expected generalization error $\Delta$ satisfies:*

$$\Delta \leq \gamma(m, \mathbb{S}) \cdot \underbrace{\mathbb{E}_i[\mathcal{L}(f_\Theta(\overline{\mathbf{h}}_i), y_i)]}_{\text{Task consistency}} + \omega(m, \mathbb{S}) \cdot \underbrace{\mathbb{E}_{(i,j) \in \mathcal{E}} \|\overline{\mathbf{h}}_i - \overline{\mathbf{h}}_j\|^2}_{\text{Topological distinguishability}}, \tag{9}$$

*where $\gamma(m, \mathbb{S}), \omega(m, \mathbb{S}) > 0$ are coefficients depending on the number of perturbed features $m$ and the selection strategy $\mathbb{S}$, and $\overline{\mathbf{h}}$ is the aggregated representation. See Appendix C.3 for more details.*

**Corollary 1.** *If all nodes use the same fixed subset $\mathcal{S}$ in Alg. 1, then: $\|\overline{\mathbf{h}}_i - \overline{\mathbf{h}}_j\|^2 \approx 0, \forall (i, j) \in \mathcal{E}$, and the propagation operator (e.g., GCN) reduces to mean-pooling, weakening structural discrimination.*

**Privacy Analysis.** A total of two rounds of $\epsilon/2$-perturbation based on $\Pi$ are sequentially applied in `TOGL`, yielding an overall guarantee of $\epsilon$-LDP by the Theorem 1. Furthermore, by the Theorem 2, the subsequent GNN training does not degrade the privacy guarantee. See Appendix C.4 for more details.

## 5 EXPERIMENTS

We conduct a series of experiments to evaluate the effectiveness of our method. Sec. 5.1 details the experimental setup, while Sec. 5.2 reports and analyzes the results in detail. Additional results on extended datasets and ablations are provided in Appendix D.5, including evaluations on large-scale datasets, robustness under noisy or sparse labels, robustness under structural privacy, and empirical resistance to inference attacks, among others.

### 5.1 EXPERIMENTAL SETTINGS

**Datasets.** We conduct extensive experiments on six representative real-world datasets spanning two domains: *citation networks* (Cora, Citeseer, and Pubmed (Yang et al., 2016)) and *social networks* (LastFM (Rozemberczki & Sarkar, 2020), Twitch (Rozemberczki et al., 2021), and Facebook (Rozemberczki et al., 2021)). The key statistics of these datasets are summarized in Table 1. Refer to Appendix D.1 for more details.

Table 1: Statistics of datasets.

| Type | Dataset | Nodes | Edges | Features | Classes |
|---|---|---|---|---|---|
| Citation Network | Cora | 2,708 | 5,278 | 1,433 | 7 |
| | Citeseer | 3,327 | 4,552 | 3,703 | 6 |
| | Pubmed | 19,717 | 44,324 | 500 | 3 |
| Social Network | LastFM | 7,624 | 27,806 | 7,842 | 18 |
| | Twitch | 4,648 | 61,706 | 128 | 2 |
| | Facebook | 22,470 | 170,912 | 4,714 | 4 |

**GNN Models.** We consider seven representative GNN models: GCN (Kipf & Welling, 2017), Graph-SAGE (Hamilton et al., 2017), GAT (Velickovic et al., 2018), GIN (Xu et al., 2019), APPNP (Klicpera et al., 2019), SGC (Wu et al., 2019), and SSGC (Zhu & Koniusz, 2021). Each model consists of two graph convolution layers with 64 neurons in each hidden layer, utilizing ReLU as the activation function (Klambauer et al., 2017) and dropout (Baldi & Sadowski, 2013) for regularization. Please refer to Appendix D.2 for more details. By default, GCN is used as the backbone model.

**LDP Mechanisms.** We consider six LDP mechanisms for protecting node features. Among them, three state-of-the-art (SOTA) baselines: the piecewise mechanism (PM) (Pei et al., 2023; Wang et al., 2019a), the multi-bit mechanism (MB) (Sajadmanesh & Gatica-Perez, 2021; Lin et al., 2022; Jin & Chen, 2022), and the square wave mechanism (SW) (Li et al., 2024; 2020). The other three are classical baselines: the 1-bit mechanism (1B) (Ding et al., 2017), the Laplace mechanism (LP) (Phan et al., 2017), and the Analytic Gaussian mechanism (AG) (Balle & Wang, 2018). These mechanisms inject random noise into the original node features based on a predefined privacy budget $\epsilon > 0$, providing formal LDP guarantees. All six mechanisms are implemented independently according to their original formulations, not by modifying any specific baseline framework. The three SOTA mechanisms (PM, MB, SW) are unified under the general LDP perturbation protocol $\Pi$ presented in Alg. 1, while the three classical mechanisms are implemented following their standard definitions in Appendix D.3. For clarity, LPGNN refers specifically to the framework proposed by Sajadmanesh & Gatica-Perez (2021) using the MB mechanism, which is included as one of our baselines. Unless

otherwise specified, `PM` is used as the default LDP perturbation mechanism. In Figures 4, 5, and 6, "SOTA" specifically refers to `PM`, which generally achieves the best or near-best performance among the three state-of-the-art mechanisms (`PM`, `MB`, `SW`) across most experimental settings.

**Parameter Settings.** For all datasets, we randomly split the nodes into training, validation, and test sets using a 50%/25%/25% ratio. Regarding the privacy budget $\epsilon$, note that the perturbation size $m$ for the three LDP mechanisms (`PM`, `MB`, and `SW`) is constrained by a coefficient $\delta$ (see Sec. 3.1 for details). To ensure $m > 1$ for meaningful evaluation, we set $\epsilon$ to values in the set $\{5.0, 7.5, 10.0, 12.5, 15.0\}$, corresponding to $m \in \{2, 3, 4, 5, 6\}$, respectively. For experiments and discussions where $m = 1$ (*i.e.*, $0 < \epsilon < 5.0$), refer to Appendix D.5. The parameter $\rho$ is selected from the set $\{0, 0.3, 0.5, 0.7, 1.0\}$, and the denoising aggregation parameter $K$ is varied over $\{0, 1, 2, 3, 4, 5\}$. Unless otherwise specified, the default $\epsilon$ is set to 10.0. Additional hyperparameter configurations are provided in Appendix D.4.

**Evaluation Metrics.** We conduct experiments on two fundamental and widely adopted tasks in graph learning: *node classification (NC)* and *link prediction (LP)*. These two tasks serve as the cornerstone of numerous downstream applications and are the primary benchmarks for evaluating graph representation methods. For evaluation, we use *classification accuracy* on the test set for the NC task, and *the area under the ROC curve (AUC)* for the LP task. Accuracy and AUC are standard and widely adopted metrics that effectively reflect model performance in NC and ranking tasks, respectively. The default task is node classification. We report the mean performance and 95% confidence intervals over 10 independent runs, calculated using bootstrapping with 1000 resamples.

## 5.2 Results & Discussion

**Effectiveness.** We comprehensively evaluate our method across six benchmark datasets under varying noise scales ($\epsilon$), comparing it with six baselines. As shown in Fig. 3, our approach consistently achieves higher node classification accuracy than all baselines, with particularly notable improvements over classical methods such as `1B`, `LP`, and `AG`, demonstrating enhanced task utility. Fig. 4 further confirms the generalizability of our method across various GNN architectures (Cora dataset), where it outperforms SOTA baselines. Furthermore, Fig. 5 presents link prediction results, showing that the effectiveness of our method extends beyond node classification by surpassing existing SOTA methods. Together, these experiments validate that our approach significantly enhances the utility of locally private graph learning across multiple tasks, datasets, and models.

**Scalability Evaluation.** To assess `TOGL`'s practicality on large graphs, we evaluated its runtime and memory usage on two representative large-scale datasets: *Co-Phy* (Shchur et al., 2018) and *Ogbn-arxiv* (Hu et al., 2020) (see Appendix D.5 'Scalability evaluation' for details). As reported in Table 8, `TOGL` introduces only moderate computational overhead compared with the PM baseline, demonstrating that it remains efficient and scalable for large-scale graph learning.

**Ablation Studies.** In this experiment, we assess two key components of our framework: task-relevant attribute analysis and aggregation denoising. Fig. 6 compares two attribute analysis strategies: Fisher Discriminant Analysis (FDA) and Sparse Model Attribution (SMA), with more detailed statistics provided in Table 17. Both outperform the SOTA baseline, with SMA slightly ahead. This modest advantage of SMA likely stems from its ability to capture feature interactions through learned model weights, whereas FDA treats each dimension independently. However, the performance gap between FDA and SMA remains small, indicating that `TOGL` is not highly sensitive to the choice of attribution method. Both successfully identify task-relevant attributes that improve utility compared to random selection, demonstrating that our framework's effectiveness stems from the general principle of task-oriented selection rather than reliance on a specific attribution technique. Fig. 7 evaluates the effect of the aggregation parameter $K$. Without aggregation ($K = 0$), utility is limited. Moderate aggregation (*e.g.*, $K = 3$ for Cora dataset) boosts performance by improving the quality of selected features. However, large $K$ values degrade utility due to over-smoothing, where node representations lose distinction. Overall, these results underscore the importance of both components and highlight the need to balance aggregation and analysis strategy for optimal privacy-utility trade-offs.

**Parameter Analysis.** In this experiment, we examine how the privacy budget $\epsilon$ and the task-topology trade-off parameter $\rho$ affect model performance. Fig. 8 shows that increasing $\epsilon$ steadily improves accuracy, as weaker noise preserves more informative features, enhancing utility. Fig. 9 explores the impact of $\rho$. Performance first rises then falls with increasing $\rho$, indicating the need for balance: low $\rho$ weakens task relevance due to random feature selection, while high $\rho$ reduces structural diversity. Further analysis is available in Appendix D.5 (Analysis of the Parameter $\rho$). For real-world

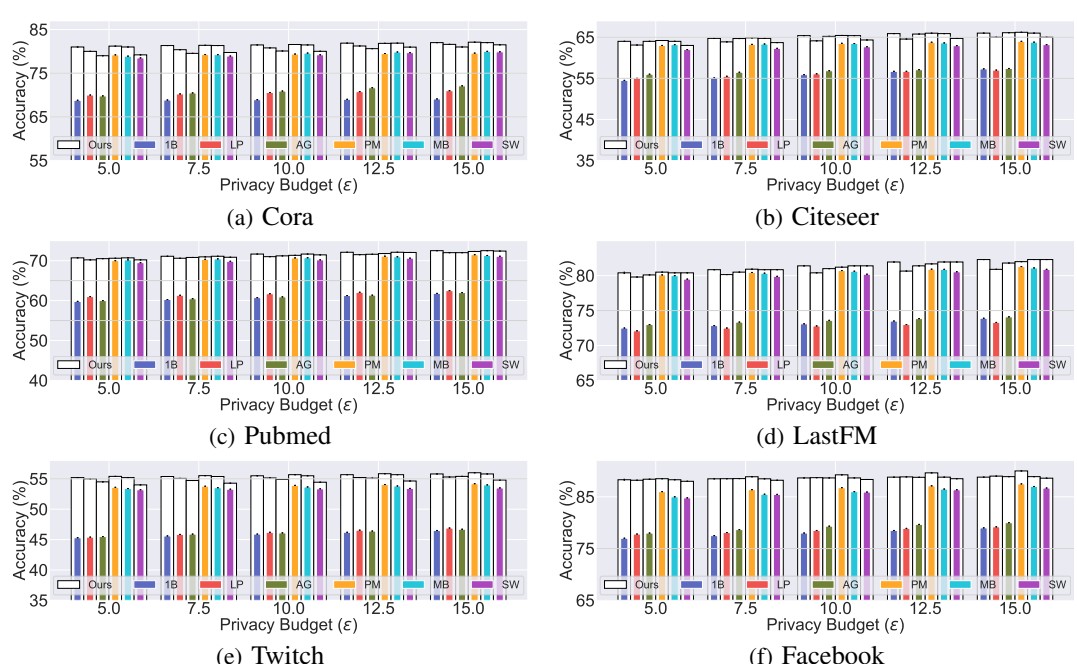

Figure 3: Performance comparison between our proposed TOGL (Ours) and existing baselines on the node classification task. The x-axis denotes the privacy budget $\epsilon$, and the y-axis indicates the test accuracy (%). Our method consistently improves the performance of all LDP mechanisms across different privacy levels. Please see Appendix D.5 for more experimental results and analysis.

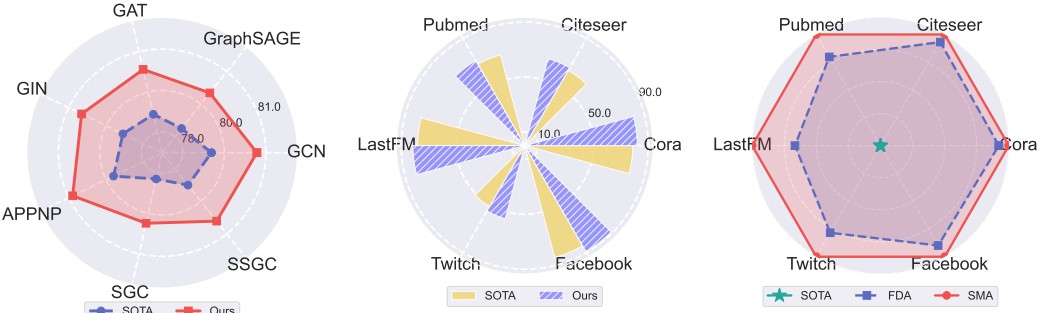

Figure 4: Comparison of accuracy (%) between our method and the SOTA baseline under different GNN models. The results show that our approach consistently outperforms the existing SOTA baseline across all cases.

Figure 5: Comparison of AUC (%) between our method and the SOTA baseline on the link prediction (LP) task. The results indicate that our method consistently achieves superior performance across all datasets.

Figure 6: Comparison of normalized accuracy $(0 \sim 1)$ using two our proposed task-relevant attribute analysis methods, FDA and SMA, within our framework across different datasets, in comparison with the SOTA baseline.

deployment, the optimal $\rho$ values typically lie within $[0.3, 0.7]$ across our experiments, reflecting an effective balance between task consistency and topological distinguishability. We recommend using $\rho = 0.5$ as a robust default choice, which consistently achieves near-optimal or optimal performance across all six datasets without requiring validation-based tuning. When validation data is available and computational resources permit, a lightweight grid search over $\{0.3, 0.5, 0.7\}$ can further optimize performance with minimal overhead.

## 6 RELATED WORK

This section provides a brief overview of *local differential privacy* and *locally private graph learning*. Additional details and extended discussions are available in Appendix E.

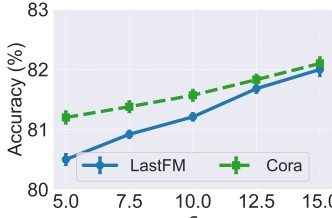 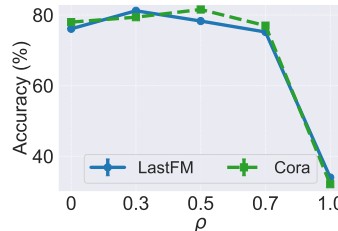

Figure 7: Comparison of model performance under varying values of the parameter $K$.

Figure 8: Comparison of model performance under varying values of the privacy budget $\epsilon$.

Figure 9: Comparison of model performance under varying values of the hyperparameter $\rho$.

**Local Differential Privacy.** LDP (Dwork et al., 2006) is a rigorous privacy notion that enables users to perturb their data locally before sharing it, thereby eliminating the need for a trusted aggregator. Due to its strong privacy guarantees, LDP has been widely adopted in diverse data collection and analysis scenarios (Jia & Gong, 2019; Li et al., 2020; Wang et al., 2017; 2019a; Asi et al., 2022).

**Locally Private Graph Learning.** Recently, locally private graph learning (Sajadmanesh & Gatica-Perez, 2021; Lin et al., 2022; Pei et al., 2023; Li et al., 2024; He et al., 2025a; Jin & Chen, 2022) has emerged as a promising research area within the privacy and security community. To support this paradigm, researchers have developed several mechanisms for perturbing node features under LDP constraints. For instance, (Sajadmanesh & Gatica-Perez, 2021) extended the one-bit mechanism (Ding et al., 2017) to high-dimensional node features via the multi-bit (`MB`) mechanism. Follow-up work proposed the piecewise (`PM`) (Pei et al., 2023) and square wave (`SW`) (Li et al., 2024) mechanisms to further enhance utility. While these methods have demonstrated effectiveness in tasks such as node classification (Kipf & Welling, 2017; Lin et al., 2022), their utility remains limited. To address this challenge, we propose a task-oriented framework for locally private graph learning. To the best of our knowledge, this is the first work to incorporate task-awareness into LDP-constrained graph learning, significantly enhancing utility while preserving strong privacy guarantees.

## 7 CONCLUSION

In this work, we propose `TOGL`, a novel task-oriented framework for locally private graph learning. The framework operates in three phases: ① *locally private feature perturbation*, ② *task-relevant attribute analysis*, and ③ *task-oriented private learning*. This structured design enables the explicit identification of feature dimensions most relevant to the downstream task prior to perturbation, thereby maximizing the retention of informative signals while ensuring strong local differential privacy guarantees. Extensive experiments on six representative real-world graph datasets demonstrate that `TOGL` consistently outperforms existing methods in terms of both utility and privacy preservation. For a discussion of broader impact, limitations, and the use of LLMs, please refer to Appendix G.

## ETHICS STATEMENT

This work fully complies with the ICLR Code of Ethics. It raises no ethical concerns: all experiments are conducted on publicly available benchmark datasets (*e.g.*, Cora (Yang et al., 2016)) that contain no personally identifiable information. No human subjects, sensitive attributes, or private data beyond these open datasets are involved. Our proposed methods are designed to enhance privacy-preserving graph learning under LDP, thereby strengthening privacy guarantees rather than introducing risks.

## REPRODUCIBILITY STATEMENT

We have taken several measures to ensure the reproducibility of our work. ① The proposed framework, algorithms, and theoretical analyses are described in detail in Sections 3 and 4, with complete proofs provided in Appendix C. ② Experimental settings, including GNN models, LDP mechanisms, and hyperparameters, are documented in Section 5, Appendix B, and Appendix D. ③ All benchmark datasets (*e.g.*, Cora) used in our experiments are publicly available, and we provide detailed descriptions in Appendix D.1. ④ An anonymous implementation of our framework is included in the supplementary material to facilitate replication of our results.

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

# A NOTATIONS

We summarize the important notations of our paper in Table 2.

Table 2: Notations.

| Notation | Description |
|---|---|
| $\mathcal{V}, \mathcal{E}, \mathbf{X}, \mathcal{Y}$ | Node set $\mathcal{V}$, edge set $\mathcal{E}$, feature matrix $\mathbf{X}$, label set $\mathcal{Y} = \{y_1, y_2, \cdots, y_C\}$ |
| $\mathcal{G}$ | A graph defined on $\mathcal{V}, \mathcal{E}, \mathbf{X}$, and $\mathcal{Y}$ |
| $\mathbf{x}_v$ | The original feature vector of user $v \in \mathcal{V}$ |
| $y_v$ | Label of node $v \in \mathcal{V}$ |
| $\Pi$ | LDP perturbation protocol for node features |
| $\mathcal{M}, \mathcal{A}, \mathcal{F}$ | LDP mechanism |
| $d$ | The number of feature dimensions |
| $m$ | The number of sampled dimensions |
| $\epsilon$ | Privacy budget |
| $\mathcal{B}$ | Perturbed feature scale |
| $\mathbf{x}'_v$ | The perturbed feature vector of user $v$ |
| $\mathcal{N}(v)$ | The set of neighbors of $v$ |
| $\mathrm{AGGREGATE}(\cdot)$ | The aggregation function |
| $\mathrm{UPDATE}(\cdot)$ | The update function |
| $\mathbf{h}_v^k$ | Original node embedding of node $v$ after $k$ aggregation steps |
| $\overline{\mathbf{h}}_v^k$ | Estimated node embedding of node $v$ after $k$ aggregation steps |
| $K$ | Aggregation denoising step |
| $\mathcal{S}$ | Set of perturbed attributes |
| $\mathcal{S}^\star$ | Set of task-relevant attributes |
| $\rho$ | Task-topology trade-off parameter |
| $\mathbb{E}[\cdot]$ | Expectation |

# B SOTA LDP MECHANISMS

In Sec. 2.3, we introduce three state-of-the-art (SOTA) LDP mechanisms designed to protect node features: the piecewise mechanism (PM) (Pei et al., 2023; Wang et al., 2019a), the multi-bit mechanism (MB) (Lin et al., 2022; Sajadmanesh & Gatica-Perez, 2021; Jin & Chen, 2022), and the square wave mechanism (SW) (Li et al., 2024; 2020). Below, we provide detailed formulations for MB and SW[3]:

- MB: The one-dimensional multi-bit mechanism perturbs a real-valued input $x \in [\alpha, \beta]$ by randomly outputting a binary value $x' \in \{-1, 1\}$ according to the following probability distribution function:

$$\Pr[x' = c \mid x] = \begin{cases} \frac{1}{e^\epsilon+1} + \frac{x-\alpha}{\beta-\alpha} \cdot \frac{e^\epsilon-1}{e^\epsilon+1}, & \text{if } c = 1 \\ \frac{e^\epsilon}{e^\epsilon+1} - \frac{x-\alpha}{\beta-\alpha} \cdot \frac{e^\epsilon-1}{e^\epsilon+1}, & \text{if } c = -1 \end{cases}. \tag{10}$$

  This mechanism achieves $\epsilon$-LDP while preserving the relative position of the input through a probabilistic encoding over two discrete values.

- SW: The one-dimensional square wave mechanism perturbs original input $x \in [\alpha, \beta]$ within an expanded domain $[-b-1, b+1]$, where the parameter $b$ is defined as:

$$b = \frac{\epsilon e^\epsilon - e^\epsilon + 1}{e^\epsilon(e^\epsilon - \epsilon - 1)}. \tag{11}$$

  The perturbed output $x'$ is then sampled from the following distribution:

$$\Pr[x' = c \mid x] = \begin{cases} p, & \text{if } c \in [x-b, x+b] \\ p/e^\epsilon, & \text{if } c \in [-b-1, x-b) \cup (x+b, b+1] \end{cases}, \tag{12}$$

---

[3]PM, MB, and SW are pure LDP mechanisms, *i.e.*, they satisfy $\epsilon$-LDP and therefore do not involve a $(\epsilon, \delta)$-style guarantee.

where $p = \frac{e^\epsilon}{2be^\epsilon + 2}$. This mechanism constructs a localized uniform distribution centered at $x$, with exponentially decaying probability for values farther from $x$, thereby balancing privacy and estimation accuracy.

## C    PROOFS & THEORETICAL ANALYSIS

### C.1    PROOF OF PROPOSITION 1

*Proof.* We consider a $d$-dimensional feature vector $\mathbf{x}_i$ for each node $v_i$, and assume that each attribute $\mathbf{x}_{i,j}, j \in [d]$ is perturbed independently using the PM mechanism. (The proof process for the other mechanisms follows the same procedure.) Let $\mathbf{x}'_{i,j}$ be the perturbed value and define the discrepancy along dimension $j$ for node $v_i$ as: $\mathbf{z}_{i,j} := \mathbf{x}'_{i,j} - \mathbf{x}_{i,j}$. According to PM, each $\mathbf{z}_{i,j}$ satisfies:

$$|\mathbf{z}_{i,j}| \leq \frac{d}{m} \cdot \frac{e^{\epsilon/2} + 1}{e^{\epsilon/2} - 1}. \tag{13}$$

Let the neighborhood of $v_i$ be $\mathcal{N}(v_i)$. The true and perturbed mean embeddings in dimension $j$ are:

$$\mathbf{h}_{i,j} := \frac{1}{|\mathcal{N}(v_i)|} \sum_{u \in \mathcal{N}(v_i)} \mathbf{x}_{u,j}, \quad \overline{\mathbf{h}}_{i,j} := \frac{1}{|\mathcal{N}(v_i)|} \sum_{u \in \mathcal{N}(v_i)} \mathbf{x}'_{u,j}. \tag{14}$$

Considering Eqs. (13), (14) and $\mathbb{E}[\mathbf{z}_{i,j}] = 0$, and apply Bernstein's inequality (Mhammedi et al., 2019), we have:

$$\Pr\left[\left|\overline{\mathbf{h}}_{i,j} - \mathbf{h}_{i,j}\right| \geq \lambda\right] = \Pr\left[\left|\sum_{i=1}^{|\mathcal{N}(v)|} \left\{\mathbf{x}'_{i,j} - \mathbf{x}_{i,j}\right\}\right| \geq |\mathcal{N}(v)| \cdot \lambda\right] \tag{15}$$

$$\leq 2 \cdot \exp\left\{-\frac{(|\mathcal{N}(v)| \cdot \lambda)^2}{2\sum_{i=1}^{|\mathcal{N}(v)|}\mathrm{Var}[\mathbf{x}'_{i,j}] + \frac{2}{3} \cdot |\mathcal{N}(v)| \cdot \lambda \cdot \frac{d}{m} \cdot \frac{e^{\epsilon/2m}+1}{e^{\epsilon/2m}-1}}\right\}. \tag{16}$$

Then, we can obtain:

$$\Pr\left[\left|\overline{\mathbf{h}}_{i,j} - \mathbf{h}_{i,j}\right| \geq \lambda\right] \leq 2 \cdot \exp\left\{-\frac{(\lambda|\mathcal{N}(v)|)^2}{\mathcal{O}(\frac{md}{\epsilon^2}) + \lambda\mathcal{O}(\frac{d}{\epsilon})}\right\}. \tag{17}$$

By applying the union bound, we have:

$$\Pr\left[\max_{j \in \{1,\dots,d\}} \left|\overline{\mathbf{h}}_{i,j} - \mathbf{h}_{i,j}\right| \geq \lambda\right] = \bigcup_{j=1}^{d} \Pr\left[\left|\overline{\mathbf{h}}_{i,j} - \mathbf{h}_{i,j}\right| \geq \lambda\right] \tag{18}$$

$$\leq \sum_{j=1}^{d} \Pr\left[\left|\overline{\mathbf{h}}_{i,j} - \mathbf{h}_{i,j}\right| \geq \lambda\right] = 2d \cdot \exp\left\{-\frac{\lambda^2|\mathcal{N}(v)|}{\mathcal{O}(\frac{md}{\epsilon^2}) + \lambda\mathcal{O}(\frac{d}{\epsilon})}\right\}. \tag{19}$$

To ensure that $\max_{j \in \{1,\dots,d\}} \left|\overline{\mathbf{h}}_{i,j} - \mathbf{h}_{i,j}\right| < \lambda$ holds with at least $1 - \varrho$ probability, it is sufficient to set

$$\varrho = 2d \cdot \exp\left\{-\frac{\lambda^2|\mathcal{N}(v)|}{\mathcal{O}(\frac{md}{\epsilon^2}) + \lambda\mathcal{O}(\frac{d}{\epsilon})}\right\}. \tag{20}$$

Solving the above for $\lambda$, we get:

$$\lambda = \mathcal{O}\left(\frac{\sqrt{d\log(d/\varrho)}}{\epsilon\sqrt{|\mathcal{N}(v)|}}\right), \tag{21}$$

which proves that: $\left\|\Upsilon\left\langle\overline{\mathbf{h}}_{\mathcal{N}(v)} - \mathbf{h}_{\mathcal{N}(v)}\right\rangle\right\| \propto 1/|\mathcal{N}(v)|^{1/2}$, we conservatively upper bound it by $1/|\mathcal{N}(v)|$. This characterization indicates that increasing the neighborhood size $|\mathcal{N}(v)|$ reduces the discrepancy with the true aggregated node embedding, thereby calibrating the estimation error introduced by local perturbation. $\square$

## C.2 COMPLEXITY ANALYSIS

The overall computational complexity of our method primarily arises from the $K$-hop neighborhood aggregation in Phase I and the attribute relevance analysis in Phase II. Specifically, the aggregation step costs $\mathcal{O}(K \cdot |\mathcal{E}| \cdot d)$, where $|\mathcal{E}|$ is the number of edges and $d$ is the feature dimension. For attribute analysis, FDA requires $\mathcal{O}(|\mathcal{V}| \cdot d + d \log d)$, while SMA requires $\mathcal{O}(|\mathcal{V}| \cdot d \cdot I + d \log d)$, where $I$ is the number of training iterations of the sparse logistic regression. The computational complexity of TOGL scales linearly with both the graph size and feature dimensionality, ensuring that it remains practical and scalable for large-scale graphs with high-dimensional data. Furthermore, common acceleration strategies such as graph pruning (Yu et al., 2022; Liu et al., 2023) and GPU-based sparse matrix operations (Lee et al., 2020) can be directly applied to further reduce runtime overhead.

## C.3 PROOF OF THEOREM 3

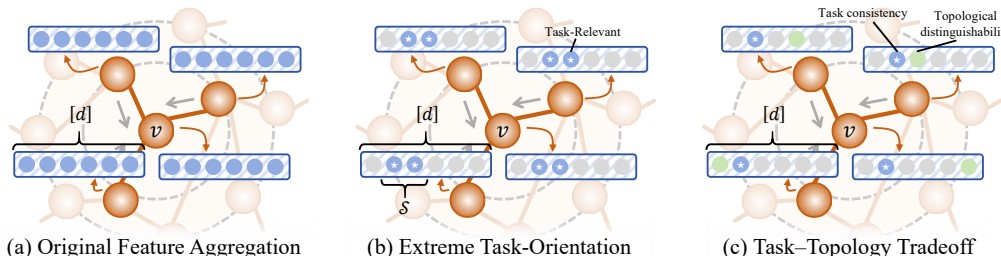

(a) Original Feature Aggregation     (b) Extreme Task-Orientation     (c) Task–Topology Tradeoff

Figure 10: Examples of Aggregation Strategies. Fig. (a) illustrates the original aggregation process without perturbation. Fig. (b) shows an extreme task-oriented aggregation strategy, which emphasizes task-relevant attributes but compromises the overall utility of graph learning due to the loss of topological distinguishability. Fig. (c) presents a trade-off example between task and topology.

*Proof.* Let $f_\Theta : \mathbb{R}^d \to \mathbb{R}^C$ be a GCN-like classifier trained under LDP-perturbed inputs $\{\mathbf{x}'_v\}_{v \in \mathcal{V}}$, where each $\mathbf{x}'_v$ is obtained by retaining only a subset of dimensions $\mathcal{S} \subset [d]$. Let $\overline{\mathbf{h}}_v \in \mathbb{R}^d$ denote the output representation of node $v$ after $K$-hop message passing.

Our goal is to bound the generalization error

$$\Delta := \mathbb{E}_i[\mathcal{L}(f_\Theta(\overline{\mathbf{h}}_i), y_i)] - \mathbb{E}_i[\mathcal{L}(f^*(\mathbf{h}_i^*), y_i)], \tag{22}$$

where $f^*$ is the oracle classifier on clean embeddings $\mathbf{h}_i^*$. Applying the triangle inequality and assuming Lipschitz continuity of $\mathcal{L} \circ f$, we have

$$\Delta \le \mathbb{E}_i[\mathcal{L}(f_\Theta(\overline{\mathbf{h}}_i), y_i)] + L_f \cdot \mathbb{E}_i\|\overline{\mathbf{h}}_i - \mathbf{h}_i^*\|, \tag{23}$$

where $L_f$ is the Lipschitz constant.

Under GCN propagation

$$\mathbf{H}^{(k)} = \sigma(\mathbf{A}\mathbf{H}^{(k-1)}\mathbf{W}^{(k)}), \quad \mathbf{H}^{(0)} = \mathbf{X}', \tag{24}$$

the node embeddings gradually converge. Since all perturbed features $\mathbf{x}'_v$ share the same support $\mathcal{S}$, the variance among neighbors reduces. We define the Dirichlet energy

$$\Phi(\mathbf{H}^{(K)}) := \sum_{(i,j) \in \mathcal{E}} \|\overline{\mathbf{h}}_i - \overline{\mathbf{h}}_j\|^2, \tag{25}$$

which captures topological distinguishability. When $\mathbf{x}'_v$ are restricted to the same $\mathcal{S}$, the aggregated embeddings collapse: $\Phi(\mathbf{H}^{(K)}) \to 0$.

We introduce coefficients $\gamma(m, \mathbb{S})$ and $\omega(m, \mathbb{S})$ to explicitly reflect the impact of the number of perturbed dimensions $m = |\mathcal{S}|$ and the feature selection strategy $\mathbb{S}$. Specifically, $\gamma(m, \mathbb{S})$ increases when excessive noise or poor selection reduces signal quality, and $\omega(m, \mathbb{S})$ increases with greater inter-node variance from task-aware selections, capturing enhanced topological distinguishability. Then we have

$$\mathbb{E}_i\|\overline{\mathbf{h}}_i - \mathbf{h}_i^*\|^2 \le \omega(m, \mathbb{S}) \cdot \Phi(\mathbf{H}^{(K)}), \tag{26}$$

absorbing remaining constants into $\gamma(m, \mathbb{S})$. Consequently,

$$\Delta \leq \gamma(m, \mathbb{S}) \cdot \underbrace{\mathbb{E}_i[\mathcal{L}(f_\Theta(\overline{\mathbf{h}}_i), y_i)]}_{\text{task consistency}} + \omega(m, \mathbb{S}) \cdot \underbrace{\mathbb{E}_{(i,j) \in \mathcal{E}} \|\overline{\mathbf{h}}_i - \overline{\mathbf{h}}_j\|^2}_{\text{topological distinguishability}} . \tag{27}$$

This bound shows that both the number of perturbed features $m$ and the selection strategy $\mathbb{S}$ directly influence the trade-off between task consistency and topological distinguishability. $\qquad\square$

Figure 10(a) illustrates the original feature aggregation process, while Figure 10(c) presents an example of aggregation under a task-topology trade-off. In addition, we have the following corollary:

**Corollary 2.** *If all nodes use the same fixed subset $\mathcal{S}$ in Alg. 1, then: $\|\overline{\mathbf{h}}_i - \overline{\mathbf{h}}_j\|^2 \approx 0, \forall(i, j) \in \mathcal{E}$, and the propagation operator (e.g., GCN) reduces to mean-pooling, weakening structural discrimination.*

*Proof.* Let $\mathbf{x}'_v \in \mathbb{R}^d$ be the perturbed node feature vector for node $v \in \mathcal{V}$, where only dimensions in the fixed subset $\mathcal{S} \subset [d]$ are perturbed and retained, and all others are zeroed out (as depicted in Figure 10(b)):

$$\mathbf{x}'_v[j] = \begin{cases} \text{LDP-perturbed}(x_v[j]), & \text{if } j \in \mathcal{S}, \\ 0, & \text{otherwise.} \end{cases} \tag{28}$$

Let $\mathbf{H}^{(0)} = \mathbf{X}' \in \mathbb{R}^{|\mathcal{V}| \times d}$ be the feature matrix, and consider a GCN propagation rule:

$$\mathbf{H}^{(k+1)} = \sigma\left(\hat{\mathbf{A}} \mathbf{H}^{(k)} \mathbf{W}^{(k)}\right), \quad \text{with } \mathbf{H}^{(0)} = \mathbf{X}', \tag{29}$$

where $\hat{\mathbf{A}}$ is the normalized adjacency matrix and $\sigma$ is an activation function.

Since all feature vectors $\mathbf{x}'_v$ lie in the same $|\mathcal{S}|$-dimensional subspace of $\mathbb{R}^d$, the *initial variance between nodes is significantly reduced*. When the same propagation operator is applied uniformly over this low-variance input, the representations $\mathbf{H}^{(k)}$ begin to *converge across neighbors*:

$$\|\mathbf{H}_i^{(k)} - \mathbf{H}_j^{(k)}\| \to 0, \quad \text{as } k \text{ increases, } (i, j) \in \mathcal{E}. \tag{30}$$

This phenomenon is known as over-smoothing (Keriven, 2022), and it occurs when graph convolution reduces inter-node variance due to repeated mixing over similar signals. As a result, in the limit, the GCN essentially performs an average over identical feature subspaces:

$$\mathbf{H}^{(K)} \approx \hat{\mathbf{A}}^K \mathbf{X}' \approx \mathbf{1}\mathbf{u}^\top, \tag{31}$$

*i.e.*, a *rank-one representation*, which is equivalent to *mean pooling*. Hence, the model loses its ability to discriminate structurally different nodes.

$\qquad\square$

## C.4 PRIVACY ANALYSIS

In `TOGL`, a total of two sequential perturbation steps (defined as $\Pi_1$ and $\Pi_2$) are performed under the LDP perturbation protocol $\Pi$, each satisfying $\epsilon/2$-LDP. By the sequential composition theorem of local differential privacy (Theorem 4), the overall privacy guarantee is bounded by $\epsilon$-LDP. Furthermore, as shown in Theorem 5, the subsequent GNN training phase operates solely on perturbed data and involves no further access to raw features, thus preserving the established $\epsilon$-LDP guarantee throughout the entire pipeline, formalized as:

$$\Pi_1 \text{ and } \Pi_2 \text{ are each } \epsilon/2\text{-LDP} \Rightarrow \Pi_2 \circ \Pi_1 \text{ is } \epsilon\text{-LDP}. \tag{32}$$

**Theorem 4** (Sequential Composition). *If $\mathcal{M}_i : \mathcal{X} \mapsto \mathcal{Z}_i$ satisfies $\epsilon_i$-LDP for each $i \in \{1, 2, \ldots, n\}$, then the composed mechanism $\mathcal{M} = (\mathcal{M}_1, \mathcal{M}_2, \ldots, \mathcal{M}_n) : \mathcal{X} \mapsto \prod_{i=1}^n \mathcal{Z}_i$ satisfies $(\sum_{i=1}^n \epsilon_i)$-LDP.*

**Theorem 5** (Post-Processing Invariance). *Let $\mathcal{A} : \mathcal{X} \mapsto \mathcal{Z}$ satisfies $\epsilon$-LDP, and let $\mathcal{F} : \mathcal{Z} \mapsto \mathcal{Z}'$ be any (possibly randomized) mapping. Then the composed mechanism $\mathcal{A} \circ \mathcal{F} : \mathcal{X} \mapsto \mathcal{Z}'$ satisfies $\epsilon$-LDP.*

## D    MORE DETAILS ON THE EXPERIMENTS

### D.1    DATASETS

We evaluate our proposed method TOGL on six widely used real-world graph datasets (The key statistics of these datasets are summarized in Table 1), three *citation networks* (Cora, Citeseer, and Pubmed) and three *social networks* (LastFM, Twitch, and Facebook), as described below:

- *Cora* (Yang et al., 2016): The Cora dataset is a citation network where each node represents a scientific publication and each edge indicates a citation between two papers. Node features are constructed from paper contents using a bag-of-words model, and the task is to classify papers into one of seven categories.

- *Citeseer* (Yang et al., 2016): Similar to Cora, Citeseer is another citation network consisting of scientific articles. Each article is described by a sparse word vector, and the objective is to classify the publications into one of six classes. Compared to Cora, Citeseer has a more sparse feature matrix and a less connected graph structure.

- *Pubmed* (Yang et al., 2016): The Pubmed dataset is a large-scale citation graph in the biomedical domain. Each node corresponds to a scientific paper, represented by TF-IDF weighted word vectors from the abstract, and the goal is to categorize papers into one of three medical topics. It contains significantly more nodes and edges than Cora and Citeseer.

- *LastFM* (Rozemberczki & Sarkar, 2020): LastFM is a user-user interaction graph derived from the Last.fm[4] music platform. Each node represents a user, and edges denote social relationships. Node features are based on music listening histories, and labels indicate user groups based on country. This dataset presents a more realistic social recommendation scenario.

- *Twitch* (Rozemberczki et al., 2021): The Twitch dataset is collected from the Twitch[5] streaming platform. Nodes correspond to users, and edges indicate mutual follows. Features are extracted from user activities and preferences, and the classification task typically involves predicting user affiliations such as language or game preference.

- *Facebook* (Rozemberczki et al., 2021): This dataset represents anonymized ego-networks from Facebook[6], where each node is a user and edges represent friendships. Node features are derived from user profiles, and the classification task involves predicting user categories based on social behavior. The graph is large and dense, making it suitable for evaluating scalability.

### D.2    GNN MODELS

To evaluate the effectiveness and generalizability of our method, we consider seven representative GNN models: GCN (Kipf & Welling, 2017), GraphSAGE (Hamilton et al., 2017), GAT (Velickovic et al., 2018), GIN (Xu et al., 2019), APPNP (Klicpera et al., 2019), SGC (Wu et al., 2019), and SSGC (Zhu & Koniusz, 2021). Each model consists of two graph convolution layers with 64 neurons in each hidden layer, utilizing ReLU as the activation function Klambauer et al. (2017) and dropout Baldi & Sadowski (2013) for regularization. The GAT model employs four parallel attention heads. For APPNP, SGC, and SSGC, we follow their original designs, using fixed propagation steps and linear classifiers. All models are implemented in PyTorch using the PyTorch-Geometric (PyG) library[7]. The experiments are carried out on a server running Ubuntu 22.04 LTS, equipped with dual Intel® Xeon® Gold 6348 CPUs, 100 GB RAM, and an NVIDIA® A800 GPU. Specifically as follows:

- *GCN*[8] (Kipf & Welling, 2017): Graph Convolutional Networks (GCNs) are a foundational GNN model that performs neighborhood aggregation via spectral graph convolutions. It introduces layer-wise propagation to aggregate and transform node features from adjacent nodes using the normalized graph Laplacian.

---

[4]https://www.last.fm/

[5]https://www.twitch.tv/

[6]https://www.facebook.com/

[7]https://www.pyg.org

[8]https://pytorch-geometric.readthedocs.io/en/latest/generated/torch_geometric.nn.models.GCN.html

- *GraphSAGE*[9] (Hamilton et al., 2017): Graph Sample and Aggregate (GraphSAGE) is an inductive GNN that learns node embeddings by sampling and aggregating feature information from local neighborhoods. It supports various aggregator functions such as mean, LSTM, or pooling, making it more flexible for large-scale and dynamic graphs.

- *GAT*[10] (Velickovic et al., 2018): Graph Attention Networks (GATs) enhance message passing by introducing attention mechanisms that learn the importance of neighboring nodes. This allows for adaptive weighting of neighbors during aggregation and improves performance in graphs with noisy or unbalanced neighborhoods.

- *GIN*[11] (Xu et al., 2019): Graph Isomorphism Networks (GINs) are designed to maximally distinguish graph structures and are proven to be as powerful as the Weisfeiler-Lehman graph isomorphism test. GINs use MLPs and sum aggregation to capture rich structural information.

- *APPNP*[12] (Klicpera et al., 2019): Approximate Personalized Propagation of Neural Predictions (APPNP) decouples feature transformation and propagation using personalized PageRank (Gleich, 2015). It first applies a shallow neural network for feature transformation, followed by multiple propagation steps that enhance long-range information flow while mitigating oversmoothing.

- *SGC*[13] (Wu et al., 2019): Simple Graph Convolution (SGC) simplifies the GCN by removing nonlinearities and collapsing weight matrices between layers. This leads to a linear and more efficient model, while retaining competitive performance.

- *SSGC*[14] (Zhu & Koniusz, 2021): Simplified Spatial Graph Convolution (SSGC) enhances SGC (Wu et al., 2019) by combining it with residual connections and spatial message passing. It introduces a tunable residual propagation mechanism that strengthens representation learning across multiple hops, improving performance while retaining the efficiency and simplicity of linear models.

### D.3 CLASSICAL LDP MECHANISMS

We consider three widely adopted classical LDP mechanisms in our framework and baseline comparisons (Section 5.1): the *1-bit mechanism* (1B) (Ding et al., 2017), *Laplace mechanism* (LP) (Phan et al., 2017), and *Analytic Gaussian mechanism* (AG) (Balle & Wang, 2018). These mechanisms are commonly used for perturbing scalar or vector-valued data under strong local privacy guarantees.

- *1-bit Mechanism* (Sajadmanesh & Gatica-Perez, 2021). The 1-bit mechanism perturbs a one-dimensional input $x \in [\alpha, \beta]$ by mapping it probabilistically to either $+1$ or $-1$, using an encoding based on the input value and privacy budget $\epsilon$. Specifically, the output $x' \in \{-1, 1\}$ is sampled as:

$$\Pr[x' = c | x] = \begin{cases} \frac{1}{e^\epsilon + 1} + \frac{x - \alpha}{\beta - \alpha} \cdot \frac{e^\epsilon - 1}{e^\epsilon + 1}, & \text{if } c = 1 \\ \frac{e^\epsilon}{e^\epsilon + 1} - \frac{x - \alpha}{\beta - \alpha} \cdot \frac{e^\epsilon - 1}{e^\epsilon + 1}, & \text{if } c = -1 \end{cases}. \tag{33}$$

- *Laplace Mechanism* (Phan et al., 2017). The Laplace mechanism adds noise sampled from a Laplace distribution to the original input. For a scalar value $x \in [\alpha, \beta]$, the perturbed value is:

$$x' = x + \text{Lap}(2/\epsilon), \tag{34}$$

where $\text{Lap}(b)$ denotes a Laplace distribution with scale $b$.

- *Analytic Gaussian Mechanism* (Balle & Wang, 2018). The Analytic Gaussian mechanism introduces noise drawn from a calibrated Gaussian distribution. For input $x \in [\alpha, \beta]$, the output is:

$$x' = x + \mathcal{N}(0, \sigma^2), \tag{35}$$

where $\sigma$ is chosen based on the privacy parameters $\epsilon$ and $\delta$ using the analytic calibration procedure described in. For example:

$$\sigma^2 = \frac{2 \ln(1.25/\delta)}{\epsilon^2} \tag{36}$$

ensures that the mechanism satisfies $(\epsilon, \delta)$-LDP. Compared to Laplace noise, the AG mechanism provides tighter tail bounds and better utility under high-dimensional settings.

---

[9]https://pytorch-geometric.readthedocs.io/en/latest/generated/torch_geometric.nn.models.GraphSAGE.html
[10]https://pytorch-geometric.readthedocs.io/en/latest/generated/torch_geometric.nn.conv.GATConv.html
[11]https://pytorch-geometric.readthedocs.io/en/latest/generated/torch_geometric.nn.conv.GINConv.html
[12]https://pytorch-geometric.readthedocs.io/en/latest/generated/torch_geometric.nn.conv.APPNP.html
[13]https://pytorch-geometric.readthedocs.io/en/latest/generated/torch_geometric.nn.conv.SGConv.html
[14]https://pytorch-geometric.readthedocs.io/en/latest/generated/torch_geometric.nn.conv.SSGConv.html

For a $d$-dimensional vector, these mechanisms apply perturbation to each dimension with a privacy budget of $\epsilon/d$, thereby ensuring overall $\epsilon$-LDP.

### D.4 PARAMETER SETTINGS

For all datasets, we randomly split the nodes into training set, validation set, and test set using a 50%/25%/25% ratio. Regarding the privacy budget $\epsilon$, note that the perturbation size $m$ for the three LDP mechanisms (PM, MB, and SW) is constrained by a coefficient $\delta$. To ensure $m > 1$ for meaningful evaluation, we set $\epsilon$ to values in the set $\{5.0, 7.5, 10.0, 12.5, 15.0\}$, corresponding to $m \in \{2, 3, 4, 5, 6\}$, respectively. The parameter $\rho$ is selected from the set $\{0, 0.3, 0.5, 0.7, 1.0\}$, and the denoising aggregation parameter $K$ is varied over $\{0, 1, 2, 3, 4, 5\}$. Unless otherwise specified, the default $\epsilon$ is set to 10.0. We perform hyperparameter tuning via grid search. The learning rate is drawn from $\{10^{-1}, 10^{-2}, 10^{-3}\}$, weight decay from $\{0, 10^{-5}, 10^{-4}, 10^{-3}\}$, and dropout rate from $\{0, 10^{-3}, 10^{-2}, 10^{-1}\}$. All models are optimized with Adam (Kingma & Ba, 2014) for a maximum of 300 epochs. The model yielding the lowest validation loss is selected for final evaluation.

### D.5 MORE EXPERIMENTAL RESULTS

To provide a more comprehensive evaluation of TOGL, we conducted additional experiments that examine its robustness, scalability, privacy protection, etc. The results are summarized as follows.

**Effect of small privacy budgets.** When the privacy budget is small ($0 < \epsilon < 5$), the local perturbation mechanism restricts the perturbation size to $m = 1$ due to the influence of the coefficient $\delta$. In this setting, we consider two extreme perturbation strategies: ① the task-consistent (TC) extreme, where all nodes perturb the same task-relevant attribute, and ② the topology-distinguishability (TD) extreme, where each node independently perturbs a random dimension from the full feature space $[d]$, regardless of task relevance. As shown in Fig. 3, the latter strategy surprisingly yields significantly higher accuracy under tight privacy constraints. This phenomenon arises because enforcing task consistency across all nodes collapses the feature diversity in the graph: since every node shares the same perturbed dimension, their representations become indistinguishable after message passing. In contrast, the random strategy—while individually suboptimal—preserves sufficient variance across the graph, maintaining topological distinguishability that benefits the downstream GNN learning process.

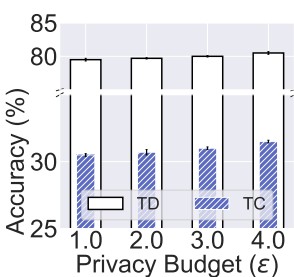

Table 3: Performance comparison between the task-consistency (TC) extreme and the topology-distinguishability (TD) extreme (Cora dataset, PM mechanism).

**Robustness under noisy or sparse labels.** We examined TOGL's performance when node labels are privatized for protection. Specifically, each label $y \in \{1, \ldots, C\}$ is perturbed via randomized response (Kairouz et al., 2016), where the privatized label $\tilde{y}$ follows:

$$\mathbb{P}[\tilde{y} = y] = \frac{e^{\epsilon_{\text{label}}}}{e^{\epsilon_{\text{label}}} + C - 1}, \quad \mathbb{P}[\tilde{y} = y'] = \frac{1}{e^{\epsilon_{\text{label}}} + C - 1}, \quad \forall y' \neq y, \tag{37}$$

with $\epsilon_{\text{label}}$ controlling the privacy level. To mitigate excessive noise, we apply Drop training (Sajadmanesh & Gatica-Perez, 2021) to smooth labels across the graph before task-relevant attribute analysis. As shown in Table 4, TOGL consistently outperforms the baseline (PM) even under noisy supervision ($\epsilon_{\text{label}} = 1.0$). Furthermore, Table 5 demonstrates that TOGL maintains robustness across varying label privacy levels $\epsilon_{\text{label}} \in \{1.0, 2.0, 3.0, \infty\}$, with only gradual degradation as $\epsilon_{\text{label}}$ decreases.

Table 4: Performance under label privacy.

| Method | Cora | LastFM |
|---|---|---|
| Baseline | 75.9 | 76.2 |
| TOGL | **78.2** | **78.9** |

Table 5: Impact of varying $\epsilon_{\text{label}}$ on Cora.

| $\epsilon_{\text{label}}$ | 1.0 | 2.0 | 3.0 | $\infty$ |
|---|---|---|---|---|
| Baseline | 75.9 | 77.3 | 78.1 | 79.4 |
| TOGL | **78.2** | **80.1** | **80.5** | **81.6** |

**Fairness analysis.** Beyond utility and privacy, an important concern in practical deployment is whether the proposed perturbation mechanism introduces unintended biases across different groups.

To investigate this, we conduct a fairness analysis. Since datasets such as Cora do not contain explicit sensitive attributes (*e.g.*, gender or race), we adopt *node degree* as a proxy sensitive feature, which is commonly regarded as a structural indicator of potential group disparity. We then compute the Pearson correlation (Benesty et al., 2009) between `TOGL`'s top-5 selected attributes and node degree for both `TOGL` and the baseline (`PM`). As shown in Table 6, both methods maintain similarly low correlations (all below 0.1). These results indicate that task-oriented attribute selection does not introduce additional fairness concerns compared to random selection, as neither approach systematically favors high-degree or low-degree nodes, thereby mitigating potential structural bias.

**Scalability evaluation.** In addition to accuracy and fairness, a key requirement for practical deployment is scalability: the method should remain efficient when applied to large graphs. To evaluate this, we conducted experiments on two representative large-scale datasets, covering both feature-rich graphs (*Co-Phy* (Shchur et al., 2018)) and large-scale citation networks (*Ogbn-arxiv* (Hu et al., 2020)). Dataset statistics are summarized in Table 7. Both `TOGL` and the `PM` baseline were evaluated under identical experimental settings on a machine equipped with dual Intel® Xeon® Gold 6348 CPUs, 100 GB RAM, and an NVIDIA® A800 GPU. The runtime and peak GPU memory usage are reported in Table 8. As shown, `TOGL` introduces only moderate computational overhead compared with the baseline, confirming that it remains practical for large-scale graph learning.

Table 6: Correlation between selected attributes and node degree.

| Dataset | Baseline | TOGL |
|---|---|---|
| Cora | 0.08 | 0.07 |
| Pubmed | 0.09 | 0.09 |

Table 7: Statistics of large-scale datasets used for scalability evaluation.

| Dataset | Nodes | Edges | Features | Classes |
|---|---|---|---|---|
| Co-Phy (Shchur et al., 2018) | 34,493 | 247,962 | 8,415 | 5 |
| Ogbn-arxiv (Hu et al., 2020) | 169,343 | 1,166,243 | 128 | 40 |

Table 8: Runtime and memory overhead on large-scale datasets.

| Dataset | Method | Runtime (s) | Peak GPU (GB) |
|---|---|---|---|
| Co-Phy | Baseline | 21 | 1.2 |
| | TOGL | 24 | 1.3 |
| Ogbn-arxiv | Baseline | 57 | 2.1 |
| | TOGL | 64 | 2.4 |

**Stability of attribute selection.** To assess the stability of `TOGL`'s attribute selection, we ran the method 10 times on Cora and measured the Jaccard similarity (Niwattanakul et al., 2013) of the top-5 selected attributes as well as the variance in classification accuracy. Results in Table 9 show high consistency (0.78 Jaccard) and low performance variance ($\pm 0.3\%$). These results indicate that the selected attribute subsets are highly consistent, and that small variations in selection do not lead to significant drops in performance, confirming the robustness of `TOGL`'s attribute selection process.

Table 9: Stability analysis on Cora.

| Metric | Value |
|---|---|
| Avg. Jaccard similarity | 0.78 |
| Accuracy (mean $\pm$ std) | $81.2 \pm 0.3$ % |

**Attribute-level sensitivity.** To gain deeper insights into `TOGL`'s per-attribute behavior, we examined both task utility and privacy protection at the attribute level on the Cora with $\epsilon = 5.0$. Specifically, for each attribute, we measured: ① *classification accuracy* (CA%) obtained when the attribute is used in the task, and ② *attribute inference attack accuracy* (AA%) on the perturbed attribute. We also recorded whether each attribute was selected by `TOGL`'s task-oriented selection module. Table 10 summarizes the results. We observe that `TOGL` preferentially selects attributes that contribute more to task utility after perturbation (*e.g.*, A5), while all attributes maintain similarly low attack accuracy, indicating that per-attribute privacy protection is consistently enforced. These findings confirm that `TOGL` effectively balances utility and privacy: it prioritizes attributes with high task relevance without compromising the rigorous LDP guarantees applied uniformly across all features.

**Empirical resistance to inference attacks.** To empirically evaluate `TOGL`'s privacy protection beyond its formal LDP guarantee, we conducted a preliminary attribute inference attack experiment (Meng et al., 2023). In this setting, an attacker observes the perturbed features of a target

node's neighbors and attempts to infer the target's sensitive attribute via majority voting, leveraging homophily (McPherson et al., 2001) in the graph. We compared TOGL (with $\epsilon = 5.0$) against a non-private baseline (NonPriv) on two representative datasets, Cora and LastFM. Attack accuracy (%) is reported in Table 11, where lower values indicate stronger privacy protection. As shown, TOGL reduces the attacker's success rate to below 20% on both datasets, representing a substantial improvement over the non-private setting, where attack accuracy exceeds 90%. These results demonstrate that TOGL provides strong empirical resistance to attribute inference attacks, complementing its formal LDP guarantees and confirming its practical privacy effectiveness.

Table 10: Attribute-level sensitivity analysis.

| Attribute | CA (%) | AA (%) | Selected |
|---|---|---|---|
| A5 | 81.2 | 17.9 | ✓ |
| A14 | 79.3 | 18.1 | ✗ |

Table 11: Attribute inference attack accuracy (%).

| Method | Cora | LastFM |
|---|---|---|
| NonPriv | 97.2 | 96.5 |
| TOGL | **18.7** | **17.5** |

**Generalizability to different LDP variants.** TOGL is designed to be compatible with a wide range of LDP mechanisms. Beyond the six classical feature-level LDP mechanisms evaluated in Sec. 5.1, we further tested TOGL under two additional variants: *Condensed Local Differential Privacy* (CLDP) (Gursoy et al., 2019; Zhang et al., 2025) and *Personalized Local Differential Privacy* (PLDP) (Li et al., 2022c; He et al., 2025c) on the Cora dataset. Table 12 reports the results, showing that TOGL consistently outperforms the corresponding baselines (PM) across both variants. This demonstrates that TOGL generalizes well to diverse LDP paradigms, including classical and modern variants, making it a flexible and broadly applicable framework for privacy-preserving graph learning.

Table 12: Performance of TOGL under different LDP variants on Cora.

| Variant | Baseline | TOGL |
|---|---|---|
| CLDP | 79.8 | **81.9** |
| PLDP | 80.6 | **82.7** |

Table 13: Multi-task performance of TOGL on Cora dataset ($\epsilon = 5.0$).

| Method | Classification ACC | Regression MAE |
|---|---|---|
| Baseline | 78.7 | 0.184 |
| TOGL | **81.5** | **0.142** |

**Robustness under structure privacy.** While this paper primarily focuses on protecting users' node features, which are often the most sensitive, our approach is orthogonal to existing privacy-preserving techniques for neighbor lists (Hidano & Murakami, 2024; Zhu et al., 2023a) and can be seamlessly integrated with them. To demonstrate this compatibility, we combined TOGL with the BLINK mechanism (Zhu et al., 2023a), which enforces link-level LDP via Bayesian estimation (Kruschke, 2013). Specifically, for a node $v \in \mathcal{V}$, let $\mathcal{N}(v)$ denote its true set of neighbors. Under link-level LDP with privacy budget $\epsilon_{\text{link}}$, each potential edge $(v, u)$ is independently perturbed:

$$\tilde{A}_{vu} = \begin{cases} 1, & \text{with probability } \frac{e^{\epsilon_{\text{link}}}}{e^{\epsilon_{\text{link}}}+1}, \\ 0, & \text{otherwise,} \end{cases} \quad \forall u \in \mathcal{N}(v), \tag{38}$$

where $\tilde{A}_{vu}$ is the perturbed adjacency entry. Non-neighbor edges are perturbed similarly, ensuring $\epsilon_{\text{link}}$-LDP for all links. The perturbed graph $\tilde{G}$ is then used as input to TOGL's task-oriented attribute selection and perturbation pipeline, leaving the rest of the method unchanged.

We conducted experiments on Cora and LastFM with $\epsilon_{\text{feature}} = 5.0$ and $\epsilon_{\text{link}} = 5.0$. Table 14 reports classification accuracy. As shown, TOGL consistently outperforms the baseline even under combined feature and structure privacy, confirming that the method remains effective when neighbor lists are privatized. This demonstrates that TOGL's contributions are *not diminished by structural privacy concerns*, but rather focus on a complementary and practically critical dimension of graph privacy.

**Analysis of the Parameter** $\rho$. As discussed in Section 5.2, Figure 9 illustrates a clear non-monotonic trend: performance peaks at moderate $\rho$ values (0.3 or 0.5), but drops dramatically as $\rho \to 1$, while degrading more mildly as $\rho \to 0$. This asymmetry arises because exclusively selecting task-relevant attributes ($\rho = 1$) causes all nodes to perturb the same fixed dimensions $\mathcal{S}^\star$, which eliminates topological distinguishability and leads to severe over-smoothing. As proven in Corollary 1, this causes graph convolution to degenerate into mean-pooling, losing structural discrimination capability

and resulting in the dramatic drop to approximately 30% accuracy. In contrast, when $\rho$ is small, the selected attributes, though random and less task-aligned, still preserve structural diversity across nodes, which helps stabilize GNN training and maintain moderate utility despite weaker task signals. Selecting an intermediate $\rho$ balances these effects, retaining enough task-relevant signal while preserving structural diversity, thereby maximizing overall utility.

**Multi-task learning with task-specific attributes.** TOGL can naturally accommodate multi-task learning scenarios by assigning task-specific attribution scores to each feature and aggregating them before feature selection and perturbation. Its fully pluggable and differentiable attribution modules (FDA and SMA) allow independent evaluation for each task. To validate this capability, we conducted a multi-task experiment on Cora, where each node is associated with: ① a

Table 14: Classification accuracy (%) under combined feature and link-level LDP ($\epsilon_{\text{feature}} = 5.0$, $\epsilon_{\text{link}} = 5.0$).

| Method | Cora | LastFM |
|---|---|---|
| Baseline + BLINK | 76.5 | 75.9 |
| TOGL + BLINK | **79.8** | **78.6** |

classification task predicting the paper's category, and ② a regression task predicting the $\ell_2$-norm of the node's feature vector, serving as a proxy for content complexity. Feature attribution scores were computed separately for each task and combined via weighted averaging. Results under $\epsilon = 5.0$ using GCN with separate task heads are summarized in Table 13. TOGL achieves superior performance on both tasks, demonstrating its flexibility in handling conflicting or partially overlapping task-specific attributes and validating its generality in multi-task private learning.

**Effect of $K$-hop denoising on baseline and TOGL.** To examine whether incorporating $K$-hop denoising into baseline methods alters the comparative performance, we conducted experiments applying the same denoising procedure ($K' = 3$) to both TOGL and the PM baseline. The denoised variants are denoted TOGL* and Baseline*, respectively. Table 15 and Table 16 report the classification accuracy (%) on Cora and LastFM. While denoising improves performance for both methods, TOGL* consistently outperforms Baseline*, demonstrating that task-oriented perturbation provides additional utility beyond standard denoising. These results highlight the complementary benefits of TOGL's selective attribute perturbation and graph-aware denoising.

Table 15: Classification accuracy (%) without denoising.

| Method | Cora | LastFM |
|---|---|---|
| Baseline | 79.4 | 80.7 |
| TOGL | 81.6 | 81.2 |

Table 16: Classification accuracy (%) with $K$-hop denoising ($K' = 3$).

| Method | Cora | LastFM |
|---|---|---|
| Baseline* | 83.7 | 85.6 |
| TOGL* | **85.1** | **86.3** |

Table 17: Detailed ablation study results comparing FDA and SMA methods with the SOTA baseline (PM). All values are classification accuracy (%) on the test set.

| Method | Cora | Citeseer | Pubmed | LastFM | Twitch | Facebook |
|---|---|---|---|---|---|---|
| Baseline | 79.4 | 63.5 | 69.8 | 80.7 | 53.9 | 86.8 |
| FDA | 81.4 | 65.2 | 71.1 | 81.8 | 55.3 | 89.0 |
| SMA | **81.6** | **65.4** | **71.3** | **82.4** | **55.7** | **89.3** |

**Comparison with alternative feature selection methods.** To further validate our choice of FDA and SMA as task-relevant attribute analysis methods, we conducted additional ablation studies comparing them with three other representative feature selection algorithms commonly used in machine learning:

- *Mutual Information (MI)* (Peng et al., 2005): Measures the mutual dependence between each feature and the class labels, capturing both linear and non-linear relationships.
- *Chi-Square ($\chi^2$)* (Liu & Setiono, 1995): A statistical test that evaluates the independence between features and labels, widely used for classification tasks.
- *PCA-based selection (PCA)* (Jolliffe & Cadima, 2016): Selects features based on their contributions to the top principal components that explain the most variance.

The comparative results across all six datasets are presented in Table 18. Our methods (FDA and SMA) consistently outperform the alternative baselines across all six datasets by 1.0-1.7% on average.

This advantage stems from their design considerations for LDP-perturbed data: FDA explicitly models class separability under noise through inter-class and intra-class variance analysis, while SMA leverages task-specific model weights through sparse logistic regression. In contrast, MI and $\chi^2$ rely on statistical dependencies that can be obscured by LDP noise, and PCA prioritizes variance explanation rather than task relevance. These results validate that FDA and SMA represent well-motivated and effective choices for task-oriented attribute selection under LDP constraints.

Table 18: Comparison of feature selection methods. All values are classification accuracy (%) on the test set under $\epsilon = 10.0$ with the PM mechanism.

| Method | Cora | Citeseer | Pubmed | LastFM | Twitch | Facebook | Avg. |
|---|---|---|---|---|---|---|---|
| MI (Peng et al., 2005) | 80.3 | 64.1 | 70.5 | 81.1 | 54.3 | 87.5 | 72.97 |
| $\chi^2$ (Liu & Setiono, 1995) | 80.1 | 63.8 | 70.3 | 80.8 | 54.1 | 87.2 | 72.72 |
| PCA (Jolliffe & Cadima, 2016) | 79.8 | 63.9 | 70.1 | 80.9 | 54.0 | 87.0 | 72.62 |
| FDA (Ours) | **81.4** | **65.2** | **71.1** | **81.8** | **55.3** | **89.0** | **73.97** |
| SMA (Ours) | **81.6** | **65.4** | **71.3** | **82.4** | **55.7** | **89.3** | **74.28** |

# E    RELATED WORKS

## E.1    LOCAL DIFFERENTIAL PRIVACY

LDP (Dwork et al., 2006) is a rigorous privacy notion that enables users to perturb their data locally before sharing it, thereby eliminating the need for a trusted aggregator. Due to its strong privacy guarantees, LDP has been widely adopted in diverse data collection and analysis scenarios, including frequency estimation (Jia & Gong, 2019; Li et al., 2020; Wang et al., 2017), mean estimation (Asi et al., 2022; Ding et al., 2017; Wang et al., 2019a), heavy hitter detection (Jia & Gong, 2019; Zhu et al., 2023b; Wang et al., 2019c), and frequent itemset mining (Li et al., 2022a; Tong et al., 2024).

Beyond single-round protocols, several works have explored two-round or multi-round LDP mechanisms for various analytics tasks (Qin et al., 2017; Sun et al., 2019; Imola et al., 2021; 2022; Liu et al., 2022b; Huang et al., 2024; He et al., 2024b;a; 2025b). These protocols typically use the first round to collect noisy global statistics and the second to refine or calibrate the result. However, they are primarily designed for *aggregate statistical estimation*, and do not involve *private learning or attribute-level decision-making*. In contrast, TOGL is the first to introduce a task-oriented two-round LDP pipeline for private graph learning, where the second round performs selective perturbation of task-relevant attributes—rather than uniformly or randomly perturbing all attributes. This setting poses unique challenges: attribute selection must be conducted over noisy data, without violating local privacy guarantees, and must account for utility-preserving structure in the graph. Our modular three-phase design addresses these challenges in a principled way and extends beyond the typical two-round estimation frameworks seen in prior work.

## E.2    LOCALLY PRIVATE GRAPH LEARNING

Recently, locally private graph learning (Sajadmanesh & Gatica-Perez, 2021; Lin et al., 2022; Pei et al., 2023; Li et al., 2024; Jin & Chen, 2022) has emerged as a promising research area within the privacy and security community. To support this paradigm, researchers have developed several mechanisms for perturbing node features under LDP constraints. For instance, (Sajadmanesh & Gatica-Perez, 2021) extended the 1-bit mechanism (Ding et al., 2017) to high-dimensional node features via the multi-bit (MB) mechanism. Follow-up work proposed the piecewise (PM) (Pei et al., 2023) and square wave (SW) (Li et al., 2024) mechanisms to further enhance utility. While these methods have demonstrated effectiveness in tasks such as node classification (Kipf & Welling, 2017), their utility remains limited. To address this challenge, we propose a task-oriented framework for locally private graph learning. To the best of our knowledge, this is the first work to incorporate task-awareness into LDP-constrained graph learning, significantly enhancing utility while preserving strong privacy guarantees. In addition, some studies have also addressed the privacy protection of topological structures (Hidano & Murakami, 2024; Zhang et al., 2024a; Zhu et al., 2023a), focusing on

obfuscating edges or degree information to prevent reconstruction attacks. Our method is orthogonal to such techniques and can be seamlessly integrated.

### E.3 KEY FEATURE SELECTION

Identifying key features (Wu et al., 2021; El Akadi et al., 2008; Lu et al., 2007; Duval & Malliaros, 2021; Ying et al., 2019; Li et al., 2017) is a long-standing problem in supervised learning, especially when only a small subset of attributes contributes meaningfully to the prediction task. While extensive research has been conducted on this topic, most existing methods assume full access to clean data and are not applicable under strong privacy constraints.

In contrast, our approach focuses on identifying task-relevant attributes from LDP-perturbed data in the graph learning setting. Unlike prior feature selection methods, we must operate without access to raw features, and we explicitly consider the trade-off between task consistent and topology distinguishability during selection. To the best of our knowledge, this is the first work to integrate feature relevance estimation into locally private graph learning.

## F PRACTICAL AVAILABILITY OF TASK SIGNALS

We clarify that the "*task-specific signals*" required by TOGL are broadly defined and not limited to explicit node labels. This paper focuses on the widely adopted semi-supervised learning setting, where training a GNN already requires a subset of labeled nodes. These existing labels are fully sufficient for TOGL's Phase II feature relevance estimation, and no additional supervision is introduced. Even under stricter conditions such as label privacy or structural privacy, we evaluate TOGL in Appendix D.5 (Sections '*Robustness under noisy or sparse labels*' and '*Robustness under structure privacy*'), and the method remains effective. TOGL's design principles are also compatible with other learning paradigms such as self-supervised learning (Liu et al., 2022a), where gradients from contrastive or predictive objectives can provide proxy task signals. Extending TOGL to these settings, however, involves additional technical considerations and is thus left for future investigation.

## G LIMITATIONS & BROADER IMPACTS

### G.1 LIMITATIONS

The scope of this work focuses primarily on homophilic graphs, which represent the dominant setting in privacy-preserving graph learning. Consequently, heterophilic graphs lie outside the main scope of our current study, and the aggregation scheme in Phase I is designed with homophily assumptions in mind. We acknowledge that naïve neighborhood aggregation may be less effective on strongly heterophilic graphs, where connected nodes often belong to different classes. However, this limitation affects only the efficiency of Phase I denoising, not the validity of our overall task-oriented LDP framework, as Phases II and III remain independent of homophily assumptions and continue to provide utility gains through selective attribute perturbation.

To address potential extensions to heterophilic settings, we have conducted a thorough survey of heterophilic graph learning methods and identified several promising directions that could be incorporated into TOGL in future work. These include *higher-order neighborhood mixing* (Abu-El-Haija et al., 2019), *ego-neighbor separation* and *combination of intermediate representations* designed for heterophily (Zhu et al., 2020), *geometric convolutions* (Pei et al., 2020), and *global attention architectures* (Mostafa & Nassar, 2020). Incorporating these advanced aggregation schemes into our denoising phase could further improve TOGL's robustness across diverse graph structures and represents an important direction for future research.

In addition, our current framework is designed for static graphs. Extending TOGL to dynamic graphs (Pareja et al., 2020; Trivedi et al., 2019), where node features and graph structure evolve over time, introduces additional challenges in privacy preservation, temporal consistency, and adaptive feature selection.

### G.2 BROADER IMPACTS

This work contributes to the development of privacy-preserving graph learning by improving utility under local differential privacy constraints. It may benefit applications involving sensitive graph-structured data, such as healthcare (Li et al., 2022b) and social networks (Sankar et al., 2021), by enabling safer and more effective learning without compromising user privacy.

### G.3 USE OF LARGE LANGUAGE MODELS

LLMs were used only as assistive tools for language polishing. They did not contribute to research ideation, experimental design, or theoretical development. All scientific content, including algorithms, analyses, and results, was generated solely by the authors. The authors take full responsibility for all content, and no LLM is listed as an author.

