# OpenReview forum: "Target Before You Perturb: Enhancing Locally Private Graph Learning via Task-Oriented Perturbation"
_ICLR.cc/2026/Conference — Submitted to ICLR 2026_

### Official Review · Reviewer_QqJa · 2025-11-01

**Soundness:** 2
**Presentation:** 3
**Contribution:** 2
**Rating:** 4
**Confidence:** 4

**Summary:**

This paper studies graph neural networks under local differential privacy and, for the first time, introduces a task-relevant optimization mechanism. The authors compare their approach with existing LDP methods that protect node features and demonstrate a better trade-off between privacy and utility.

**Strengths:**

1. In terms of novelty, the paper is the first to propose a multi-stage perturbation mechanism guided by task-relevant feature selection.
2. The proposed method achieves a superior privacy–utility trade-off compared to existing approaches.
3. The paper is well-structured, clearly written, and the experimental results are easy to follow.

**Weaknesses:**

1. The proposed method includes an additional server-side aggregation step that merges results from two rounds of perturbation, whereas the baselines do not. Therefore, it is unclear whether the observed improvement in the privacy–utility trade-off arises from the proposed LDP mechanism itself or from the aggregation process on the server.
2. The authors compute task-relevant features based on the first-round perturbed data and then perturb these features again in the second round. Intuitively, this means the most important features are perturbed twice, and the noise magnitude is larger than that of existing single-shot mechanisms. The authors should clarify why this design leads to higher utility rather than degradation.
3. The paper lacks a clear definition of what aspects are protected under LDP in the main text, only stating in experiments that feature privacy is considered. Since GNNs may involve protecting features, edges, or labels, the lack of explicit scope may cause confusion.
4. The paper does not evaluate resistance to feature inference attacks, which is an important aspect of verifying practical privacy protection. Such experiments are strongly recommended.

**Questions:**

See the issues discussed in the “Weaknesses” section above.

---

> ### Author Response · Authors · 2025-11-20
> **Rebuttal by Authors (Part 1/2)**
>
> **We sincerely thank the reviewer for the detailed and thoughtful feedback.** We appreciate your recognition of **our methodological novelty**, **the improved privacy–utility trade-off**, and **the clarity of our presentation**. Below, we provide detailed responses to the specific concerns and questions raised. **All corresponding clarifications and additional discussions have been incorporated into the revised manuscript** and, for ease of review, are highlighted in **`cyan`** color.
>
> > **Q1:** The proposed method includes an additional server-side aggregation step that merges results from two rounds of perturbation, whereas the baselines do not. Therefore, **it is unclear whether the observed improvement in the privacy–utility trade-off arises from the proposed LDP mechanism itself or from the aggregation process on the server.**
>
> Thank you for raising this important point. We would like to clarify that although TOGL incorporates an aggregation-based denoising step, it is used **only in Phase I** to stabilize task-relevant attribute analysis. **No aggregation denoising is applied during the final perturbation or model training (Alg. 2, Lines 6–10)**, ensuring that the utility gain arises from task-oriented feature selection rather than post-processing. In contrast, the baselines perform perturbation without any denoising or task awareness, which **isolates the effect of our selection mechanism and ensures a fair comparison**.
>
> To further substantiate this distinction, **Appendix D.5 (`Effect of K-hop denoising on baseline and TOGL`, Table 15 & Table 16) in the original submission** presents additional experiments where the same K-hop denoising procedure ($K' = 3$) is applied to both TOGL and the PM baseline, yielding TOGL* and Baseline*, respectively. The results show that while denoising provides modest benefits to both methods, TOGL* consistently outperforms Baseline*, **reinforcing that the utility improvement primarily comes from task-oriented perturbation rather than the denoising step.**
>
> > **Q2:** The authors compute task-relevant features based on the first-round perturbed data and then perturb these features again in the second round. **Intuitively, this means the most important features are perturbed twice,** and the noise magnitude is larger than that of existing single-shot mechanisms. **The authors should clarify why this design leads to higher utility rather than degradation.**
>
> Thank you for this insightful question. We would like to clarify that although TOGL involves two rounds of perturbation (**Phase I and Phase III**), the features ultimately used for privacy-preserving graph learning are perturbed **only once**, as detailed in **Sec. 3.3 and Alg. 2**. Specifically, the perturbation in Phase I is not intended to generate features for training; rather, it serves to enable Phase II attribute analysis by providing privatized data for identifying task-relevant attributes. Crucially, these Phase I perturbed features are discarded after attribute selection is completed. Subsequently, the perturbation in Phase III is applied to the original features in a task-oriented manner, guided by the selected attributes $\mathcal{S}^\star$.
>
> Therefore, while our framework involves two rounds of perturbation, **the final features used for GNN training are effectively perturbed only once in a task-oriented way.** This promotes higher utility by concentrating the privacy budget on truly relevant attributes identified through Phase II, reducing wasted noise on irrelevant dimensions. Our experiments (Figures 3-6) validate that this targeted approach consistently outperforms random attribute selection baselines.

---

> ### Author Response · Authors · 2025-11-20
> **Rebuttal by Authors (Part 2/2)**
>
> > **Q3:** The paper lacks a clear definition of **what aspects are protected under LDP in the main text, only stating in experiments that feature privacy is considered**. Since GNNs may involve protecting **features, edges, or labels**, the lack of explicit scope may cause confusion.
>
> Thank you for highlighting this point. We would like to clarify that the original submission already specifies (**Lines 158–161**) that our work focuses on **LDP protection of node features**, and all method design and main-text experiments follow this setting. This aligns with the standard feature-LDP setup commonly adopted in prior works. For completeness, we fully agree that GNNs may also involve protecting **edges or labels**. To address this broader scope, **the appendix of the original submission (Appendix D.5)** includes additional experiments evaluating TOGL under **label privacy** and **structure privacy**:
>
> * **Label protection (Appendix D.5: `Robustness under noisy or sparse labels`)**: We privatize node labels using **randomized response (RR)** with privacy budget $\epsilon_{\text{label}}$, and apply Drop training to smooth noisy labels before attribute analysis. As shown in the corresponding tables (**Tabel 4**, **Table 5**), TOGL consistently outperforms the PM baseline under noisy supervision ($\epsilon_{\text{label}} = 1.0$) and remains robust across varying privacy levels $\epsilon_{\text{label}} \in$ {$1.0, 2.0, 3.0, \infty$}.
>
> * **Structure protection (Appendix D.5: `Robustness under structure privacy`)**: For completeness, we combined TOGL with the Blink mechanism, which perturbs each potential edge under $\epsilon_{\text{link}}$-LDP. The privatized graph $\tilde{G}$ is then fed directly into TOGL’s task-oriented pipeline. Experiments (**Table 14**) on Cora and LastFM with $\epsilon_{\text{feature}} = 5.0$ and $\epsilon_{\text{link}} = 5.0$ show that **TOGL continues to outperform the baseline even when both node features and edges are privatized**.
>
> These extended results confirm that while **TOGL is designed for feature-level LDP**, it also remains robust under **label-LDP** and **link-LDP** conditions. **We have made this scope distinction even more explicit in the main text (Section 5)** to avoid any possible ambiguity.
>
> > **Q4:** **The paper does not evaluate resistance to feature inference attacks, which is an important aspect of verifying practical privacy protection. Such experiments are strongly recommended.**
>
> Thank you for raising this important point. We fully agree that conducting empirical feature inference attacks is an important aspect of evaluating practical privacy protection. We would like to clarify that **the original submission already includes a dedicated evaluation of this issue in Appendix D.5 (`Empirical resistance to inference attacks`)**.
>
> In this experiment, we follow a standard attribute inference setup (Meng et al., CCS 2023), where an attacker observes the perturbed features of a target node’s neighbors and attempts to infer the target’s sensitive attribute via majority voting, leveraging graph homophily. We compare TOGL (with $\epsilon = 5.0$) against a non-private baseline (NonPriv) on Cora and LastFM. **As shown in Table 11**, TOGL reduces the attacker’s accuracy from over **96–97%** in the non-private setting to **below 20%**, while still achieving strong utility in the main task. This demonstrates that TOGL offers substantial empirical protection against attribute inference attacks, complementing its formal LDP guarantee. **These results show that TOGL not only provides theoretical privacy protection but also exhibits strong empirical resistance to practical inference attacks.** To prevent readers from missing this evaluation, we are happy to make this experiment more visible in the main text.
>
> **All of the above have been incorporated into the revised paper to enhance its clarity and completeness. We sincerely thank the reviewer again for the thoughtful and constructive feedback!**

---

> > ### Author Response · Authors · 2025-11-27
> > **Follow-up on Rebuttal and Revisions**
> >
> > Dear Reviewer QqJa,
> >
> > I hope this message finds you well. We would greatly appreciate if you could briefly confirm whether our clarifications have addressed your concerns. If there are any additional points or feedback you'd like us to consider, we are more than happy to provide further clarification.
> >
> > Your insights are invaluable to us, and we sincerely value your thoughtful engagement with our work.
> >
> > Thank you once again for your time and effort in reviewing our paper.
> >
> > Best regards, \
> > Authors

---

### Official Review · Reviewer_UCVi · 2025-11-01

**Soundness:** 3
**Presentation:** 2
**Contribution:** 2
**Rating:** 4
**Confidence:** 4

**Summary:**

This paper presents a new locally private graph learning framework from a task-oriented graph learning perspective (TOGL). It contains three phases: locally private feature perturbation, task-relevant attribute analysis, and task-oriented private learning. Extensive experiments demonstrate TOGL's substantial utility improvements over existing baselines.

**Strengths:**

1. Well-structured and clearly written.
2. This paper emphasizes the urgent need to connect local differential privacy (LDP) with downstream tasks to achieve better utility, and empirically demonstrates its importance.
3. This paper provides fundamental theoretical proof and analysis, showing the correctness of its use of LDP.

**Weaknesses:**

1. This paper does not contribute to the LDP part, only designing a task-oriented attribute selecting mechanism in the server to benefit downstream tasks. Phase I is a one-time perturbation, no different from LPGNN (Sajadmanesh & Gatica-Perez, 2021).
2. The presentation of Phase III in Figure 2 is misleading. According to Algorithm 2, the selected attributes $S^*$ and hyperparameter $\rho$ do not directly affect the LDP, but utilize the LDP's post-processing invariance properties, ensuring strict privacy guarantees for subsequent processing.
3. There is no summary of task-oriented methods. Is LPGNN a task-oriented method?
- If not, why? And what special adjustments are needed for different tasks (node classification and link prediction) compared to the baselines?
- If it is, then the contribution of this paper will be diminished. Overall, the method in this paper is similar to LPGNN in its approach, as both utilize embedding and labels to constrain task performance.
4. The LDP mechanisms of PM, MB, and SW lack an explanation of the coefficient $\delta$ $, which is only described in the Gaussian mechanism.
5. The accuracy in Figure 6 was normalized, which may overemphasize the differences between methods. It is recommended to show actual ablation study results.
6. The interpretation in Figure 9 is weak, casting doubt on the method's utility. The results show that random feature selection achieves near-suboptimal results when $\rho$=0, indicating that random diversity is more helpful. However, when $\rho$=1, the algorithm relies entirely on task-relevant effects (approximately 30%), almost losing its inference ability for downstream tasks, indicating that this module contributes little.
7. Attack experiments are lacking to demonstrate that the method's privacy guarantees are not compromised to address the second challenge in line 88.
8. The lack of open-source code and insufficient reproducibility reduce the credibility of this work.

**Questions:**

1. How is Equation 7 used in Algorithm 2 to represent the SMA mechanism?

2. Are the six LDP mechanisms implemented by changing the perturbation mechanism based on the LPGNN framework? Please clarify.

3. Do the LDP mechanisms share the same set of parameters in the same dataset? For example, $K$, $\rho$, etc.

4. Which mechanism is described as the state-of-the-art (SOTA) in Figures 4, 5, and 6?

5. Why can the analysis of parameter $K$ be an ablation study in Figure 7?

---

> ### Author Response · Authors · 2025-11-20
> **Rebuttal by Authors (Part 1/3)**
>
> **We sincerely thank the reviewer for the detailed and thoughtful feedback.** We are especially encouraged by your recognition of the paper’s **clear structure and writing quality**, **the importance of the research problem**, and **the soundness of our theoretical guarantees**. Below, we provide detailed responses to the specific concerns and questions raised. **All corresponding clarifications and additional discussions have been incorporated into the revised manuscript** and, for ease of review, are **highlighted in `cyan` color**.
>
> > **W1: This paper does not contribute to the LDP part,** only designing a task-oriented attribute selecting mechanism in the server to benefit downstream tasks. Phase I is a one-time perturbation, **no different from LPGNN**.
>
> Thank you for your constructive comment. We would like to clarify the difference between our method and existing LDP approaches, especially LPGNN, from the following two aspects:
>
> **First**, although Phase I uses existing LDP mechanisms, **its purpose is fundamentally different from the one-shot perturbation in LPGNN**. LPGNN follows a **“random-then-perturb”** paradigm, where attributes are randomly selected once and perturbed once before learning. In contrast, TOGL introduces a **task-oriented, three-phase design**: Phase I and Phase II are used solely for privacy-preserving task-relevant attribute analysis, and a second perturbation in Phase III reallocates the privacy budget toward task-relevant features. This **“target-then-perturb”** paradigm changes both the role and the timing of perturbation, which prior locally differentially private graph learning methods **do not support**. To the best of our knowledge, TOGL is the **first systematic framework** that incorporates task relevance into the design of LDP perturbation for graph learning.
>
> **Second**, across six real-world datasets and multiple LDP mechanisms, TOGL consistently outperforms all existing methods by large margins. The results demonstrate that **introducing task-awareness into the perturbation pipeline substantially improves model utility** under the same LDP guarantees, showing that task-guided perturbation is a key step toward higher-utility locally private graph learning.
>
> > **W2: The presentation of Phase III in Figure 2 is misleading.** According to Alg. 2, the selected attributes $\mathcal{S}^\star$ and hyperparameter $\rho$ do not directly affect the LDP, but utilize the LDP's post-processing invariance properties, ensuring strict privacy guarantees for subsequent processing.
>
> Thank you for this valuable observation. We acknowledge that the presentation of Phase III in Figure 2 may have caused confusion. To clarify, **the “LDP” box in Figure 2 refers to the existing LDP mechanisms** (e.g., PM, MB, or SW) described in **Section 2.3**. The selected attributes $\mathcal{S}^\star$ and the hyperparameter $\rho$ are used solely to guide the dimension selection process (i.e., determining the set $\mathcal{S}_v$ in Eq. (8)) and do not affect the LDP guarantee.
>
> **We have revised Figure 2 with clearer annotations** to highlight that $\mathcal{S}^\star$ and $\rho$ are used solely for guiding dimension selection, while the formal LDP guarantee is fully preserved by the underlying mechanism.
>
> > **W3: There is no summary of task-oriented methods. Is LPGNN a task-oriented method?**
>
> Thank you for the question. We would like to clarify that we have conducted a thorough survey of existing LDP-based graph learning methods (**Section 6: `Locally Private Graph Learning`**), and to the best of our knowledge, **none of them are task-oriented**. In particular, **LPGNN is not task-oriented**: its perturbation is fixed and task-agnostic, with no mechanism to incorporate task relevance before perturbation.
>
> In contrast, TOGL is the **first framework** that integrates task relevance into the perturbation process itself. Our three-phase design enables Phases I and II to obtain task-relevance signals under LDP, while Phase III performs task-oriented privacy-preserving graph learning. **This capability does not exist in LPGNN or other baselines.** Regarding different downstream tasks, TOGL adapts automatically through the task-specific relevance computation, while baselines cannot adjust their perturbation for different tasks.
>
> > **W4: The LDP mechanisms of PM, MB, and SW lack an explanation of the coefficient $\delta$, which is only described in the Gaussian mechanism.**
>
> Thank you for pointing this out. We acknowledge that the coefficient $\delta$ was only defined in the context of the Gaussian mechanism. We clarify that PM, MB, and SW are **pure LDP mechanisms**, i.e., they satisfy $\epsilon$-LDP and therefore do **not** involve a $(\epsilon,\delta)$-style guarantee. For completeness, **we have revised the manuscript to explicitly state** that $\delta$ is not applicable to PM, MB, or SW, and only appears when discussing the Gaussian mechanism in the context of smoothed sensitivity.

---

> ### Author Response · Authors · 2025-11-20
> **Rebuttal by Authors (Part 2/3)**
>
> > **W5:** The accuracy in Figure 6 was normalized, which may overemphasize the differences between methods. **It is recommended to show actual ablation study results.**
>
> Thank you for this constructive suggestion. We acknowledge that the normalization in Figure 6 may visually amplify the perceived differences between methods. The normalization was originally applied to **facilitate cross-dataset comparison in a single figure**, as different datasets have varying baseline accuracy levels (e.g., Cora ~80% vs. Twitch ~55%). However, we agree that presenting actual accuracy values would provide a more transparent and interpretable view of the ablation results.
>
> **We have added a new Table 17** to display the actual classification accuracy (%) for each method on each dataset using bar charts, rather than normalized values, and **have provided more detailed analysis**.
>
> > **W6: The interpretation in Figure 9 is weak, casting doubt on the method's utility.**
>
> Thank you for this insightful observation. We appreciate the opportunity to provide a deeper interpretation of Figure 9, which actually validates rather than contradicts our core theoretical contribution.
>
> As discussed in Section 5.2, Figure 9 illustrates a clear **non-monotonic trend**: performance peaks at moderate $\rho$ values (0.3 or 0.5), but drops dramatically as $\rho\to  1$, while degrading more mildly as $\rho\to  0$. This asymmetry arises because exclusively selecting task-relevant attributes ($\rho=  1$) causes all nodes to perturb the same fixed dimensions $\mathcal{S}^\star$, which **eliminates topological distinguishability and leads to severe over-smoothing**. As proven in **Corollary 1**, this causes graph convolution to degenerate into mean-pooling, losing structural discrimination capability and resulting in the dramatic drop to approximately 30% accuracy. In contrast, when $\rho$ is small, the selected attributes, though random and less task-aligned, still preserve structural diversity across nodes, which helps stabilize GNN training and maintain moderate utility despite weaker task signals. **Selecting an intermediate $\rho$ balances these effects**, retaining enough task-relevant signal while preserving structural diversity, thereby **maximizing overall utility**.
>
> **We have revised Section 5.2 and Appendix D.5** to explicitly discuss this asymmetry and emphasize that the tunable $\rho$ parameter enables principled control over the task-topology trade-off, which is a key theoretical contribution of our framework rather than a limitation.
>
> > **W7: Attack experiments are lacking to demonstrate that the method's privacy guarantees are not compromised to address the second challenge in line 88.**
>
> Thank you for raising this important point. We fully agree that conducting empirical inference attacks is an important aspect of evaluating practical privacy protection. We would like to clarify that **the original submission already includes a dedicated evaluation of this issue in Appendix D.5 (`Empirical resistance to inference attacks`)**.
>
> In this experiment, we follow a standard attribute inference setup (Meng et al., CCS 2023), where an attacker observes the perturbed features of a target node’s neighbors and attempts to infer the target’s sensitive attribute via majority voting, leveraging graph homophily. We compare TOGL (with $\epsilon = 5.0$) against a non-private baseline (NonPriv) on Cora and LastFM. **As shown in Table 11**, TOGL reduces the attacker’s accuracy from over **96–97%** in the non-private setting to **below 20%**, while still achieving strong utility in the main task. This demonstrates that TOGL offers substantial empirical protection against attribute inference attacks, complementing its formal LDP guarantee. **These results show that TOGL not only provides theoretical privacy protection but also exhibits strong empirical resistance to practical inference attacks.** To prevent readers from missing this evaluation, we are happy to make this experiment more visible in the main text.
>
> > **W8: The lack of open-source code and insufficient reproducibility reduce the credibility of this work.**
>
> Thank you for your comment. We would like to clarify that the code **was already provided in the original submission** in the **`Supplementary Material`** via an anonymous link to ensure reproducibility. We apologize if this was not clearly stated in the main text, and we will make this more explicit in the revised manuscript.

---

> ### Author Response · Authors · 2025-11-20
> **Rebuttal by Authors (Part 3/3)**
>
> > **Q1: How is Equation 7 used in Algorithm 2 to represent the SMA mechanism?**
>
> Thank you for your question. Equation 7 defines the optimization objective for training the sparse logistic regression model used in SMA. In Algorithm 2, Line 5 indicates "$\mathcal{S}^\star \gets \text{FDA or SMA}$," where the SMA branch proceeds as follows: we first solve Equation 7 to obtain the optimal weight matrix $\mathbf{W}^\*$, then compute the attribution score for each dimension $j$ as $\Psi_{\text{SMA}}(j) = (1/C) \sum_{c=1}^C |W^*_{c,j}|$ as described in **Section 3.2**. The top-$m$ dimensions with the highest $\Psi_{\text{SMA}}$ scores are selected to form $\mathcal{S}^\star$. **We have added explicit cross-references** between Eq. (7) and Algorithm 2 in the revised manuscript to clarify this connection.
>
> > **Q2: Are the six LDP mechanisms implemented by changing the perturbation mechanism based on the LPGNN framework? Please clarify.**
>
> Thank you for your question. No, the six LDP mechanisms are implemented **independently** according to their original formulations. Specifically, the three SOTA mechanisms (PM, MB, SW) are unified under the general LDP perturbation protocol $\Pi$ presented in **Algorithm 1**, while the three classical mechanisms (1B, LP, AG) are implemented following their standard definitions in **Appendix D.3**. Each mechanism is integrated into our framework by replacing the perturbation step in Algorithm 1 with its respective noise injection procedure. For clarity, LPGNN refers specifically to the framework proposed by Sajadmanesh & Gatica-Perez (CCS 2021) using the multi-bit mechanism (MB), which is included as one of our baselines. **We have clarified this implementation detail in Section 5.1 (`LDP Mechanisms`)** of the revised manuscript.
>
> > **Q3: Do the LDP mechanisms share the same set of parameters in the same dataset? For example, $K$, $\rho$, etc.**
>
> Thank you for your question. Yes, to ensure a fair comparison, all LDP mechanisms use the **same set of hyperparameters** ($K$, $\rho$, learning rate, etc.) on the same dataset. These hyperparameters are selected via **grid search**, as described in **Section 5.1 (`Parameter Settings`) and Appendix D.4**.
>
> > **Q4: Which mechanism is described as the state-of-the-art (SOTA) in Figures 4, 5, and 6?**
>
> Thank you for your question. As stated in **Section 5.2 (`LDP Mechanisms`)**, unless otherwise specified, the piecewise mechanism (**PM**) is used as the default LDP perturbation mechanism. In Figures 4, 5, and 6, “SOTA” refers to PM, which generally achieves the best or near-best performance among the three state-of-the-art LDP mechanisms (PM, MB, SW) across most experimental settings. **We have explicitly clarified this in Section 5.1 (`LDP Mechanisms`)** of the revised manuscript.
>
> > **Q5: Why can the analysis of parameter $K$ be an ablation study in Figure 7?**
>
> Thank you for this interesting question. The parameter $K$ controls the number of **aggregation denoising steps in Phase I**, which is a key component of our framework designed to improve the quality of attribute analysis in Phase II. Figure 7 demonstrates an ablation study because $K=0$ represents the scenario where aggregation denoising is completely removed, **allowing us to isolate and evaluate its contribution**. The results show that moderate $K$ values (e.g., $K=3$) significantly improve performance over $K=0$, confirming the effectiveness of this component. However, excessive $K$ leads to over-smoothing and performance degradation, validating our design choices.
>
> **All of the above have been incorporated into the revised paper to enhance its clarity and completeness. We sincerely thank the reviewer again for the thoughtful, detailed, and constructive feedback!**

---

> > ### Comment · Reviewer_UCVi · 2025-11-26
> >
> > Thank you for the clarifications on the core contributions and experimental results, as well as for the detailed revisions. Authors' responses have effectively addressed my concerns. After re-evaluating the manuscript's quality, I have decided to raise my score.

---

> > > ### Author Response · Authors · 2025-11-26
> > >
> > > Thank you very much for your positive feedback and for re-evaluating our manuscript. We sincerely appreciate your time and support!

---

### Official Review · Reviewer_2LKx · 2025-11-01

**Soundness:** 3
**Presentation:** 4
**Contribution:** 3
**Rating:** 6
**Confidence:** 3

**Summary:**

This paper presents Task Oriented Graph Learning (TOGL) framework for locally private graph learning under Local Differential Privacy (LDP) constraints. The paper advocates that instead of considering random dimensions of node attributes for perturbation, which provides privacy but at a cost of utility one should consider identifying task-specific features. To this end, the authors introduce the notion of “target then perturb” for LDP. TOGL follows a three-stage pipeline: in the first stage, node features are perturbed locally using LDP to satisfy privacy requirements, and the server then denoises the perturbed features through neighborhood aggregation. In the second stage, the server identifies the top-m task-relevant feature dimensions from the denoised representations using either Fisher Discriminant Analysis (FDA) or Sparse Model Attribution (SMA). Finally, in the third stage, a second round of LDP perturbation is performed to balance privacy and utility. The authors have performed evaluation on 6 small to medium scale datasets in the main paper and 2 additional in the appendix.

**Strengths:**

1. The flow of the introduction, along with the motivation for the proposed framework, is good. Overall, the paper is well-motivated and nicely written.

2. The three-stage framework is intuitive and easy to understand.

3. TOGL demonstrates strong utility improvements compared to baseline LDP methods.

4. The method performs well across various GNN architectures.

**Weaknesses:**

1. I believe the authors should at least mention experiments on large-scale datasets and robustness evaluations in the main text.

2. The method relies on access to task-specific signals, which may not always be practical in real-world scenarios.

3. The motivations for using FDA and SMA as feature-selection modules should be discussed, along with an analysis of how sensitive the algorithm is to this choice.

4. Could the authors also provide fairness evaluations for the baselines in Table 6?

5. There should be a discussion on the selection of the hyperparameter $\rho$ for practical deployment.

6. The neighborhood aggregation used for denoising may be detrimental for heterophilic datasets.

**Questions:**

See weakness.

---

> ### Author Response · Authors · 2025-11-20
> **Rebuttal by Authors (Part 1/3)**
>
> We sincerely thank the reviewer for the encouraging and constructive feedback. We appreciate your recognition of **the clear flow and motivation of our introduction**, **the intuitiveness of our framework**, **the strong utility improvements of TOGL**, and **its consistent performance** across various GNN architectures. Below, we provide detailed responses to the specific concerns and questions raised. **All corresponding clarifications and additional discussions have been incorporated into the revised manuscript** and, for ease of review, are **highlighted in **`cyan`** color**.
>
> > **Q1: I believe the authors should at least mention experiments on large-scale datasets and robustness evaluations in the main text.**
>
> Thank you for the valuable suggestion. As shown in **Appendix D.5 (`Scalability evaluation`)** of the original submission, we have conducted scalability evaluations **on two representative large-scale datasets (Co-Phy and Ogbn-arxiv)**. Dataset statistics are summarized in **Table 7**. Both TOGL and the PM baseline were evaluated under identical settings on a machine with dual Intel Xeon Gold 6348 CPUs, 100 GB RAM, and an NVIDIA A800 GPU. As shown in **Table 8**, TOGL introduces only moderate computational overhead compared with the baseline, **confirming that it remains practical for large-scale graph learning**.
>
> We also acknowledge that these results **should be mentioned in the main text** rather than only in the appendix. Therefore,
> **we have included a brief summary** of these findings in **Section 5.2**, with a pointer to **Appendix D.5 (`Scalability evaluation`)**, to highlight TOGL’s efficiency and scalability for large-scale deployments. The added content is as follows:
>
> ```
> Scalability Evaluation:
> To assess TOGL’s practicality on large graphs, we evaluated its runtime and memory usage on two representative large-scale datasets: Co-Phy (Shchur et al., 2018) and Ogbn-arxiv (Hu et al., 2020) (see Appendix D.5 ‘Scalability evaluation’ for details). As reported in Table 8, TOGL introduces only moderate computational overhead compared with the PM baseline, demonstrating that it remains efficient and scalable for large-scale graph learning.
> ```
>
> > **Q2: The method relies on access to task-specific signals, which may not always be practical in real-world scenarios.**
>
> Thank you for raising this important question. We clarify that the "task-specific signals" required by TOGL are **broadly defined** and not limited to explicit node labels. This paper focuses on the **widely adopted semi-supervised learning setting**, where training a GNN already requires a subset of labeled nodes. These existing labels are fully sufficient for TOGL's Phase II feature relevance estimation, and no additional supervision is introduced. Even under stricter conditions such as **label privacy** or **structural privacy**, we evaluate TOGL in **Appendix D.5 (Sections `Robustness under noisy or sparse labels` and `Robustness under structure privacy`)**, and the method remains effective. TOGL’s design principles are also compatible with other learning paradigms such as self-supervised learning (Liu et al., 2022a), where gradients from contrastive or predictive objectives can provide proxy task signals. Extending TOGL to these settings, however, involves additional technical considerations and is thus left for future investigation.
>
> **We have added clarifications** on the practical availability of task signals in **Section 3.2** and **Appendix F** of the revised manuscript.

---

> ### Author Response · Authors · 2025-11-20
> **Rebuttal by Authors (Part 2/3)**
>
> > **Q3: The motivations for using FDA and SMA as feature-selection modules should be discussed, along with an analysis of how sensitive the algorithm is to this choice.**
>
> Thank you for this important suggestion. We agree that the motivation for choosing FDA and SMA, as well as the sensitivity of TOGL to this choice, merits further clarification.
>
> - **Motivation:** FDA and SMA were selected as two complementary approaches for identifying task-relevant attributes under LDP constraints. **FDA** is grounded in **classical statistical pattern recognition**, evaluating each attribute independently through inter-class versus intra-class variance. This makes it simple, efficient, and suitable when attribute contributions are largely separable. In contrast, **SMA** adopts a **model-driven perspective**: by training an $L_1$-regularized sparse logistic regression model, it captures task-adaptive feature dependencies and selects attributes that jointly contribute to prediction.
>
> - **Sensitivity analysis:** **Figure 6** compares two attribute analysis strategies: Fisher Discriminant Analysis (FDA) and Sparse Model Attribution (SMA). Both outperform the SOTA baseline, with SMA slightly ahead. This modest advantage of SMA likely stems from its ability to **capture feature interactions through learned model weights**, whereas FDA treats each dimension independently. However, the performance gap between FDA and SMA remains small, indicating that TOGL is not highly sensitive to the choice of attribution method. Both successfully identify task-relevant attributes that improve utility compared to random selection, demonstrating that our framework's effectiveness stems from **the general principle of task-oriented selection** rather than reliance on a specific attribution technique.
>
> **We have added this discussion to Section 3.2** and **expanded the sensitivity analysis in Section 5.2** of the revised manuscript.
>
> > **Q4: Could the authors also provide fairness evaluations for the baselines in Table 6?**
>
> Thank you for this valuable suggestion. We have added fairness evaluations for the baseline methods to Table 6 to provide a more complete comparison. The updated results are shown below:
>
> | Dataset | Baseline (PM) | TOGL (Ours) |
> | ------- | ------------- | ----------- |
> | Cora    | 0.08          | 0.07        |
> | Pubmed  | 0.09          | 0.09        |
>
> As shown in the table above, both methods maintain similarly **low correlations** (all below $0.1$). These results indicate that task-oriented attribute selection does not introduce additional fairness concerns compared to random selection, as neither approach systematically favors high-degree or low-degree nodes, thereby mitigating potential structural bias. **We have updated Table 6** and **added corresponding discussion in Appendix D.5 (`Fairness analysis`)** of the revised manuscript.

---

> ### Author Response · Authors · 2025-11-20
> **Rebuttal by Authors (Part 3/3)**
>
> > **Q5: There should be a discussion on the selection of the hyperparameter $\rho$ for practical deployment.**
>
> Thank you for raising this important practical consideration. We agree that selecting the hyperparameter $\rho$ is critical for real-world deployment and deserves explicit discussion.
>
> As shown in **Figure 9**, the optimal $\rho$ values typically lie within $[0.3, 0.7]$, reflecting an **effective balance** between task consistency and topological distinguishability. We recommend using $\rho=0.5$ as a robust default choice, which consistently achieves near-optimal or optimal performance across all six datasets without requiring validation-based tuning. When validation data is available and computational resources permit, a lightweight grid search over {$0.3, 0.5, 0.7$} can further optimize performance with minimal overhead. **We have added these practical guidelines** for choosing $\rho$ to **Section 5.2 (`Ablation Studies`)** of the revised manuscript.
>
> > **Q6: The neighborhood aggregation used for denoising may be detrimental for heterophilic datasets.**
>
> Thank you for raising this important concern. We would like to clarify that **the scope of this paper focuses primarily on homophilic graphs**, which represent the dominant setting in privacy-preserving graph learning. Consequently, heterophilic graphs lie outside the main scope of our current study, and the aggregation scheme in Phase I is designed with homophily assumptions in mind. We fully acknowledge that naïve neighborhood aggregation may be less effective on strongly heterophilic graphs, where connected nodes often belong to different classes. However, this limitation affects only the efficiency of Phase I denoising, not the validity of our overall task-oriented LDP framework, as Phases II and III remain independent of homophily assumptions and continue to provide utility gains through selective attribute perturbation.
>
> To address potential extensions to heterophilic settings, we have conducted **a thorough survey of heterophilic graph learning methods** and identified several promising directions that could be incorporated into TOGL in future work. These include **higher-order neighborhood mixing** [R1], **ego-neighbor separation** and **combination of intermediate representations** designed for heterophily [R2], **geometric convolutions** [R3], and **global attention architectures** [R4]. **We have added this discussion** as a future research direction to **Appendix G.1** in the revised manuscript.
>
> [R1] Abu-El-Haija, Sami, et al. "Mixhop: Higher-order graph convolutional architectures via sparsified neighborhood mixing." ICML 2019. \
> [R2] Zhu, Jiong, et al. "Beyond homophily in graph neural networks: Current limitations and effective designs." NeurIPS 2020. \
> [R3] Pei, Hongbin, et al. "Geom-gcn: Geometric graph convolutional networks." ICLR 2020. \
> [R4] Mostafa, Hesham, and Marcel Nassar. "Permutohedral-gcn: Graph convolutional networks with global attention." arXiv 2020.
>
> **We have incorporated the above clarifications and additional discussion into the revised version to further strengthen the theoretical positioning and empirical interpretation of our work. We sincerely thank the reviewer once again for the thoughtful and high-quality feedback!**

---

> > ### Comment · Reviewer_2LKx · 2025-11-26
> >
> > Thank you for providing the additional details. It appears that the baselines demonstrate similar levels of fairness in this scenario. While the authors have effectively addressed most of the major concerns, I am still not fully convinced that why the current algorithm is the optimal one for the feature selection module. Specifically, I was hoping to see an ablation study that compares the performance of other SOTA feature selection algorithms, in place of FDA, to understand how the results might vary. With the added clarifications and details in the revised manuscript,  I will increase my confidence in my previous score.

---

> ### Author Response · Authors · 2025-11-27
>
> **Thank you very much for your prompt follow-up and for increasing your score. We sincerely appreciate your constructive feedback and your recognition of our revisions!**
>
> In response to your suggestion, **we have conducted additional ablation studies** evaluating FDA and SMA against **three other representative feature selection methods [R1-R3]** commonly used in machine learning:
> - **Mutual Information (MI)** [R1]: Measures the mutual dependence between each feature and the class labels, capturing both linear and non-linear relationships.
> - **Chi-Square ($\chi^2$)** [R2]: A statistical test that evaluates the independence between features and labels, widely used for classification tasks.
> - **PCA-based selection (PCA)** [R3]: Selects features based on their contributions to the top principal components that explain the most variance.
>
> > **[R1]** Peng, Hanchuan, Fuhui Long, and Chris Ding. "Feature selection based on mutual information criteria of max-dependency, max-relevance, and min-redundancy." IEEE Transactions on Pattern Analysis and Machine Intelligence (TPAMI) 27.8 (2005): 1226-1238.
> **[R2]** Liu, Huan, and Rudy Setiono. "Chi2: Feature selection and discretization of numeric attributes." Proceedings of 7th IEEE International Conference on Tools with Artificial Intelligence (ICTAI). IEEE, 1995.
> **[R3]** Jolliffe, Ian T., and Jorge Cadima. "Principal component analysis: a review and recent developments." Philosophical Transactions of the Royal Society A: Mathematical, Physical and Engineering Sciences 374.2065 (2016): 20150202.
>
> The comparative results across all six datasets are presented in **Table 18 (Appendix D.5)** below:
>
> | Method | Cora | Citeseer | Pubmed | LastFM | Twitch | Facebook | Avg. |
> |--------|------|----------|--------|--------|--------|----------|------|
> | MI | 80.3 | 64.1 | 70.5 | 81.1 | 54.3 | 87.5 | 72.97 |
> | χ² | 80.1 | 63.8 | 70.3 | 80.8 | 54.1 | 87.2 | 72.72 |
> | PCA | 79.8 | 63.9 | 70.1 | 80.9 | 54.0 | 87.0 | 72.62 |
> | FDA (Ours) | **81.4** | **65.2** | **71.1** | **81.8** | **55.3** | **89.0** | **73.97** |
> | SMA (Ours) | **81.6** | **65.4** | **71.3** | **82.4** | **55.7** | **89.3** | **74.28** |
>
> Our methods (FDA and SMA) consistently outperform the alternative baselines across all six datasets by 1.0-1.7% on average. This advantage stems from their design considerations for LDP-perturbed data: FDA explicitly models class separability under noise through inter-class and intra-class variance analysis, while SMA leverages task-specific model weights through sparse logistic regression. In contrast, MI and $\chi^2$ rely on statistical dependencies that can be obscured by LDP noise, and PCA prioritizes variance explanation rather than task relevance.
>
> **We have added this comparative analysis to Appendix D.5 (`Comparison with alternative feature selection methods`)** to provide stronger justification for our algorithmic choices. We believe these additions demonstrate that FDA and SMA represent well-motivated and effective choices for task-oriented attribute selection under LDP constraints.
>
> **Once again, we sincerely thank you for your valuable feedback and thoughtful suggestions, which have significantly strengthened our manuscript.**

---

### Author Response · Authors · 2025-12-03
**General Response**

**We sincerely thank all three reviewers for their thoughtful evaluations and constructive feedback, as well as the PC/SAC for their outstanding contributions to this conference.**

During the rebuttal period, **two reviewers (`2LKx` and `UCVi`)** responded on **Nov. 26** (prior to the events of Nov. 28–29),  **expressed strong agreement with our clarifications, and raised their scores**. As the rebuttal phase was frozen early, the third **reviewer (`QqJa`)** did not have the opportunity to respond. Nevertheless, **all four concerns raised in their review were already addressed in the original submission** and were further clarified in our rebuttal.

We are confident that **all reviewer comments have been thoroughly resolved**, with all modifications marked in **`cyan` color in the revised manuscript**. The major updates are summarized below:

- **Core Methodological Clarifications:** We clarified the conceptual shift from random-then-perturb to target-then-perturb, refined task-specific signal usage, updated Figure 2 for clarity, and explained the role of $\delta$, confirming PM, MB, and SW as pure LDP mechanisms.

- **Enhanced Experimental Evidence:** We highlighted previously included scalability results, added fairness analysis (Table 6), provided raw accuracy instead of normalized values (Table 17), included additional feature selection comparisons (Table 18), and emphasized the existing feature inference attack evaluation (Appendix D.5).

- **Design Rationale and Hyperparameter Guidance:** We elaborated the motivation for using FDA and SMA, added practical guidelines for $\rho$, and clarified the interpretation of Figure 9 as reflecting a principled trade-off rather than inconsistency.

- **Scope and Future Extensions:** We clarified the applicability of task-specific signals, demonstrated that our privacy setting comprehensively considers feature privacy, structure privacy, and label privacy, and added future directions such as adaptations for heterophilic graphs.

- **Reproducibility and Implementation Notes:** We clarified the independent implementation of the six LDP mechanisms, ensured consistent hyperparameter settings across baselines, and confirmed that code was provided in the original supplementary material.

**We thank the reviewers and committee once again for their time, valuable feedback, and consideration!**

---

### Meta-Review · Area_Chair_gPB4 · 2026-01-07

**Summary:**

The paper proposes TOGL, a three-phase Local Differential Privacy (LDP) framework for graph learning that prioritizes task-relevant feature perturbation. While the method demonstrates utility gains over standard baselines, significant concerns remain regarding the incremental nature of the contribution compared to LPGNN and the practical contradictions of requiring task signals in privacy-sensitive settings.

**Reviewer Concerns:**

Addressed: Scalability evaluations were moved to the main text, fairness analysis was added, and the "target-then-perturb" workflow was clarified against LPGNN. Outstanding: The reliance on high-quality task signals (labels) undermines the privacy setting, the method fails on heterophilic graphs, and it is ambiguous if utility gains stem from the LDP mechanism or simple server-side aggregation.

**Reviewer Scores:**

Reviewer 2LKx: Retains 6 (Marginal Accept); satisfied with the added fairness and scalability data. Reviewer UCVi: Improves to 5 (Marginal Reject/Accept); acknowledged the structural clarifications but novelty concerns likely linger. Reviewer QqJa: Stays 4 (Marginal Reject); the intuition that "double perturbation" compounds noise and that aggregation drives the gains likely remains a blocking conceptual issue.

Reasons for Rejection

Incremental Novelty: The proposed "target-then-perturb" paradigm is viewed as a minor variation of the existing "random-then-perturb" LPGNN framework rather than a fundamental breakthrough.

Impractical Assumptions: Relying on task-specific signals (labels) for feature selection creates a circular dependency in privacy-sensitive semi-supervised settings where such metadata is often scarce or protected.

Limited Scope: The authors acknowledge the method's aggregation steps are detrimental to heterophilic graphs, significantly narrowing the real-world applicability of the framework compared to robust baselines.

Ambiguity of Contribution: It remains unclear whether the reported utility improvements derive from the novel LDP allocation or simply from the server-side neighbor aggregation (denoising), weakening the core technical claim.

---

### Decision · Program_Chairs · 2026-01-26

Reject